# Enhancing Compositional Reasoning in CLIP via Reconstruction and Alignment of Text Descriptions

**Jihoon Kwon**
Seoul National University
kog0712@snu.ac.kr

**Kyle Min**[*]
Oracle
kyle.min@oracle.com

**Jy-yong Sohn**[†]
Yonsei University
jysohn1108@yonsei.ac.kr

## Abstract

Despite recent advances, vision-language models trained with standard contrastive objectives still struggle with compositional reasoning – the ability to understand structured relationships between visual and linguistic elements. This shortcoming is largely due to the tendency of the text encoder to focus on individual words rather than their relations, a limitation reinforced by contrastive training that primarily aligns words with visual objects. In this paper, we introduce *REconstruction and Alignment of text Descriptions* (READ), a fine-tuning method designed to enhance compositional reasoning by adding two auxiliary objectives to the contrastive learning: (1) a token-level *reconstruction* objective, where a frozen pre-trained decoder reconstructs alternative captions based on the embedding of the original caption; and (2) a sentence-level *alignment* objective, which explicitly aligns paraphrased sentences in the embedding space. We show that READ-CLIP, a model derived by applying the READ method to the pre-trained CLIP model, achieves the state-of-the-art performance across five major compositional reasoning benchmarks, outperforming the strongest conventional fine-tuning baseline by up to 4.1%. Furthermore, applying the READ to existing CLIP variants (including NegCLIP and FSC-CLIP) also improves performance on these benchmarks. Quantitative and qualitative analyses reveal that our proposed objectives – reconstruction and alignment – offer complementary benefits: the former encourages the encoder to capture relationships between words within a caption, while the latter ensures consistent representations for paraphrases expressed with different wording.

## 1 Introduction

Recent advances in Vision-Language Models (VLMs) have significantly enhanced the ability to align images with text descriptions [8, 67]. A key driver of this progress is contrastive pre-training, such as CLIP [45], which learns to embed images and texts into a shared multi-modal space, in a way that the distance in the embedding space represents the semantic similarity of image-text pairs. VLMs trained with this standard contrastive objective have been widely applied to diverse downstream tasks, including open-vocabulary object detection [17, 72], semantic segmentation [15, 30, 34], cross-modal retrieval [12, 36, 71], and multi-modal generation [1, 47, 53].

Despite their remarkable progress, current VLMs still face challenges with *compositional reasoning* – the ability to understand structured relationships between visual and linguistic elements [37, 56, 64]. Numerous studies have shown that VLMs commonly fail on even simple compositional tasks that humans find straightforward [11, 19, 24, 28, 40, 42, 58, 66, 69]. For instance, when given an image of a horse eating grass, VLMs often assign a higher similarity score to the incorrect caption "the grass is eating the horse" than to the correct caption "the horse is eating the grass", highlighting the limitation of the VLMs in capturing syntactic and relational structures [64]. These failures underscore the need for further research on compositional reasoning to achieve reliable and robust vision-language understanding in real-world applications [4, 9, 29, 33, 39, 56, 57, 60].

---

[*]Work partially done while at Intel Labs. [†]Corresponding author

39th Conference on Neural Information Processing Systems (NeurIPS 2025).

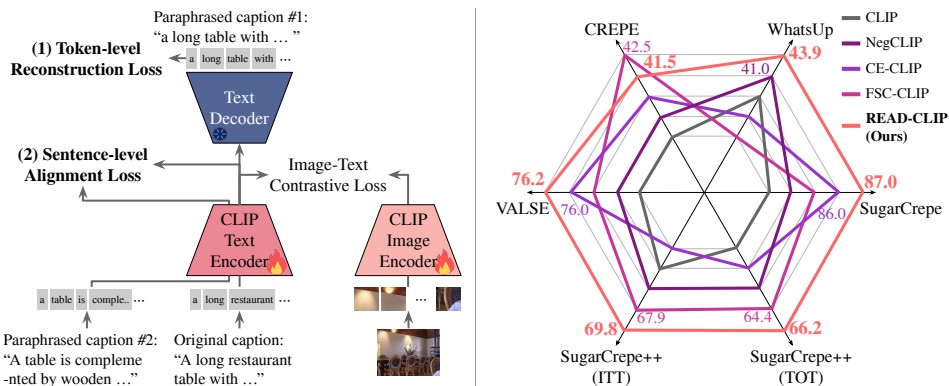

Figure 1: Overview of the training objectives used in READ-CLIP **(left)**, a VLM that applies *REconstruction and Alignment of text Descriptions* (READ) method to the pretrained CLIP model [45]. READ is our proposed fine-tuning method that enhances compositional reasoning in VLMs by augmenting contrastive learning with two auxiliary objectives. The auxiliary objectives consist of two components: *token-level reconstruction* and *sentence-level alignment*. The performance of READ-CLIP **(right)** on compositional reasoning benchmarks demonstrates that it consistently outperforms conventional state-of-the-art methods across diverse aspects of compositional reasoning.

Prior works have identified the *text encoder* as a primary bottleneck for compositional reasoning in VLMs trained with the contrastive objective. Specifically, the text encoder often fails to capture the relationship between words in a sentence [10, 23, 65]. This limitation is largely attributed to the *contrastive* objective, which trains the text encoder to align a caption with its corresponding image, without encouraging the encoder to capture the relationships between words [10, 64]. As a result, the text encoder focuses on words that refer to objects depicted in the image, as they mainly contribute to the image-text alignment, thereby limiting its ability to learn compositional reasoning [7].

Conventional approaches to overcome this limitation can be categorized into two parts. The first approach is to modify the contrastive objective by introducing *hard negatives* [48] – semantically different examples that are nonetheless difficult for the model to distinguish from the positives. Here, the model is trained to bring each sample closer to its positives and push it farther from hard negatives, thus improving the compositional reasoning capability [41, 50, 54, 64]. However, these approaches that rely solely on hard negatives often encourage the model to focus on patterns specific to those negatives, rather than developing genuine compositional reasoning [13, 14, 19]. The second approach is to add auxiliary objectives to the standard contrastive objective. However, these efforts either supervise both image and text encoders jointly [51, 70] or focus solely on the image encoder [2, 18]. Although the text encoder serves as a primary bottleneck for compositional reasoning, limited attention has been paid to adopting auxiliary objectives for the text encoder, aimed at improving the compositional reasoning capability.

In order to enhance the compositional reasoning capability of VLMs more effectively, we tackle the underexplored challenge of improving the text encoder via a targeted training objective. Specifically, our contributions are as follows:

- In Sec. 3, we propose *REconstruction and Alignment of text Descriptions* (READ), a fine-tuning method designed to enhance the text encoder by adding two auxiliary objectives to the standard contrastive objective: (1) token-level reconstruction and (2) sentence-level alignment, as in Fig. 1. First, the *token-level reconstruction* objective trains the text encoder to produce embeddings from the original caption that enable a frozen decoder to reconstruct each token of an alternative caption. Second, the *sentence-level alignment* objective explicitly aligns paraphrased captions in the embedding space to reflect their shared semantics, even when they are expressed differently.

- In Sec. 4, we provide experiments demonstrating the READ method is effective across a wide range of compositional reasoning benchmarks. Specifically, we introduce READ-CLIP, a VLM derived by applying the READ method to the pre-trained CLIP model [45], which achieves the state-of-the-art performance on five compositional reasoning benchmarks. READ-CLIP outperforms the famous baseline NegCLIP [64] by an average of 4.5% across benchmarks, and outperforms the strongest baseline FSC-CLIP [38] by up to 4.1%. Furthermore, applying

the READ method to existing CLIP variants (including NegCLIP and FSC-CLIP) consistently improves performance across these benchmarks, with gains of up to 2.4%.

- Our analysis in Sec. 5 demonstrates that the two objectives in the READ method – reconstruction and alignment – provide complementary benefits for compositional reasoning. The former encourages the encoder to capture relationships between words within a caption, while the latter ensures consistent representations for paraphrases even expressed with different wording. We also find that reconstructing an alternative caption, rather than the original caption, reduces overfitting to exact wording and improves the ability of VLMs to learn relational understanding.

## 2   Related Work

**Compositional Reasoning in Contrastive VLMs.** VLMs trained with the contrastive objective often struggle with compositional reasoning [56, 64], as the text encoder tends to overlook relationships between words due to the training objective that prioritizes image-text alignment based on object mentions [7, 10, 23, 64, 65]. To address this limitation, a common approach is to introduce hard negatives by modifying the contrastive objective. These approaches typically generate hard negative captions via rule-based perturbation [6, 64], language models [7, 68], scene graphs [20, 18, 54], or construct hard negative pairs by altering both text and image [3, 41, 51]. These methods have been shown to be effective; for example, NegCLIP improves over CLIP by 23.4% on ARO [64], and CE-CLIP [68] achieves a 7.2% gain on VALSE [40, 68]. Among these, DAC [6] highlights that training with well-aligned captions improves compositional reasoning, while TSLVC [7] finds that using paraphrased captions in analogy loss improves image classification performance.

Beyond contrastive learning, recent work has proposed adding auxiliary objectives to improve the compositional reasoning. Some methods supervise both image and text encoders, such as SF-CLIP [51], which uses masked distillation from pre-trained models, and IL-CLIP [70], which employs codebook alignment and iterative re-initialization. Other approaches target only the image encoder: SDS-CLIP [2] uses distillation from diffusion models [49], and CLIP-SGVL [18] introduces a scene-graph loss. Although the text encoder has been identified as the primary bottleneck [10, 23, 65], approaches that specifically introduce auxiliary objectives for the text encoder for improved compositional reasoning capability of VLMs remain scarce.

**Reconstruction Objectives for Training Encoders.** For the purpose of language understanding, various recent works have focused on training a text encoder-decoder architecture in a way that the sentence put into the encoder is reconstructed at the output of the decoder. It is reported that such *reconstruction objective* is beneficial for improving the performance of encoders on various language understanding benchmarks [26, 32, 59]. For instance, MASS [55] reconstructs masked fragments of the original sentence, while RetroMAE [62] reconstructs the original sentence from a pooled embedding. These approaches have shown that the auxiliary reconstruction objective can encourage the text encoder to capture both syntactic and semantic relationships among the words in the sentence [55, 62]. However, reconstructing the caption under the encoder-decoder structure has not been explored as a training objective for VLMs. Despite the use of auxiliary objectives in VLMs [31, 63], these approaches do not aim to reconstruct the input caption in the text modality. We introduce a text reconstruction loss during fine-tuning, aiming to enhance the compositional reasoning ability of VLMs.

## 3   Method

In this section, we formally define our proposed *REconstruction and Alignment of text Descriptions* (READ) method for improving the compositional reasoning performance of VLMs. The READ method is a fine-tuning method using three types of losses: a conventional *contrastive* loss reviewed in Sec. 3.1 and two auxiliary losses proposed in Sec. 3.2 and Sec. 3.3, namely, the token-level *reconstruction* loss and the sentence-level *alignment* loss. The final form of the fine-tuning loss in the READ method is given in Sec. 3.4.

### 3.1   Contrastive Loss

We consider a batch of $B$ image–text pairs, denoted as $\{(I_i, T_i)\}_{i=1}^{B}$, where $I_i$ and $T_i$ represent the $i$-th image and its associated caption. The image and text encoders are denoted by $f_I$ and $f_T$, which produce embeddings $u_i = f_I(I_i)$ and $v_i = f_T(T_i)$, respectively. For convenience, we define the index set $[B] := \{1, 2, \ldots, B\}$.

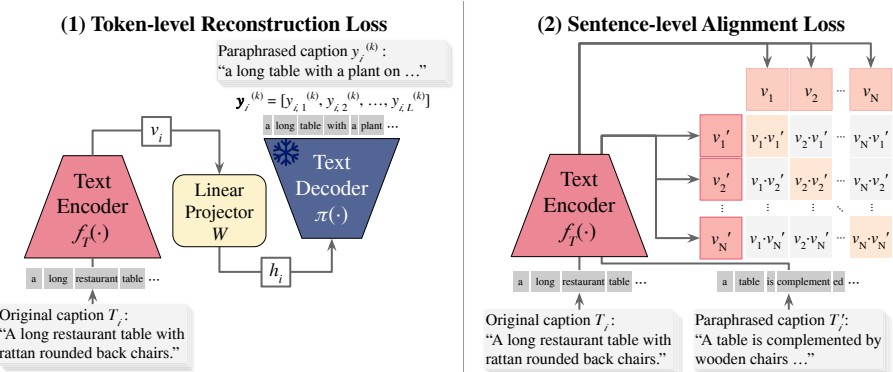

Figure 2: Illustration of our proposed auxiliary objectives of *REconstruction and Alignment of text Description* (READ) method. Given pairs of captions – an original and its paraphrase – that share a common meaning, the *token-level reconstruction* (1) trains the text encoder to produce embeddings from the original caption such that a frozen pre-trained decoder can reconstruct each token of the paraphrased caption. This reconstruction encourages the encoder to capture relationships between words within a caption, which are critical for reconstructing its paraphrase. In contrast, the *sentence-level alignment* (2) aligns the pair of captions in the embedding space. This alignment encourages the encoder to capture underlying semantic relationships across the paraphrased captions.

Suppose we are given a batch of image-text pairs $\{(I_i, T_i)\}_{i=1}^{B}$. For each $i, j \in [B]$, the similarity between the $i$-th image $I_i$ with embedding $u_i$ and the $j$-th text $T_j$ with embedding $v_j$ is defined as $\phi(I_i, T_j) := \exp\left(\cos\left(u_i, v_j\right)/\tau\right)$ where $\tau$ is a learnable temperature parameter. The standard contrastive losses used in CLIP [45] is represented as

$$\mathcal{L} = \frac{1}{2}\left(\mathcal{L}_{I \to T} + \mathcal{L}_{T \to I}\right), \tag{1}$$

where each component is defined as

$$\mathcal{L}_{I \to T} = -\frac{1}{B}\sum_{i=1}^{B}\log\frac{\phi(I_i,\ T_i)}{\sum_{j=1}^{B}\phi(I_i,\ T_j)}, \quad \mathcal{L}_{T \to I} = -\frac{1}{B}\sum_{i=1}^{B}\log\frac{\phi(T_i,\ I_i)}{\sum_{j=1}^{B}\phi(T_i,\ I_j)}, \tag{2}$$

which are dubbed as image-to-text loss and the text-to-image loss, respectively. Note that for the standard loss in Eq. 1, each image $I_i$ has one *positive* caption $T_i$ and $B-1$ *negative* captions $\{T_j\}_{j \neq i}$.

The READ method uses a variant of the standard contrastive loss in Eq. 1 to improve the compositional reasoning capability, motivated by the following two observations. First, prior works [10, 23, 65] have identified the *text encoder* as a primary bottleneck for compositional reasoning in VLMs trained with the contrastive objective. Second, recent works [41, 54, 64] showed that compositional reasoning of CLIP model is improved by introducing additional *hard negative* captions – semantically different captions that are nonetheless difficult for the model to distinguish from the positives – into the training loss. Motivated by these observations, the contrastive loss in our proposed READ method uses hard negatives in the text domain, details of which are given as below.

For each sample index $i$, let $\{\tilde{T}_i^{(m)}\}_{m=1}^{M}$ be the set of $M$ hard negative captions associated with the positive caption $T_i$. Incorporating these hard negatives into the denominator, the image-to-text loss in Eq. 2 is modified as

$$\mathcal{L}'_{I \to T} = -\frac{1}{B}\sum_{i=1}^{B}\log\frac{\phi(I_i,\ T_i)}{\sum_{j=1}^{B}[\phi(I_i,\ T_j) + \sum_{m=1}^{M}\phi(I_i,\ \tilde{T}_j^{(m)})]}. \tag{3}$$

Inserting this modified image-to-text loss in Eq. 1, the contrastive loss in the READ method is

$$\mathcal{L}_{\text{Contrastive}} = \frac{1}{2}\left(\mathcal{L}'_{I \to T} + \mathcal{L}_{T \to I}\right). \tag{4}$$

### 3.2 Token-Level Reconstruction Loss

Here we define our proposed token-level reconstruction loss and discuss how this loss promotes compositional reasoning. Given the $i$-th image–text pair $(I_i, T_i)$, we consider a set of $K$ captions

$\{\mathbf{y}_i^{(k)}\}_{k=1}^{K}$ describing the same image $I_i$, which serve as target sequences for the reconstruction. We refer to this set as the alternative captions of $I_i$. As shown in the left-hand side of Fig. 2, the token-level reconstruction loss measures how well each token in the alternative caption $\mathbf{y}_i^{(k)}$ is reconstructed at the output of the text decoder, once the original caption $T_i$ is given as the input of the text encoder.

To be specific, let $v_i = f_T(T_i)$ be the text embedding of the original caption $T_i$ for the $i$-th sample. In order to reconstruct texts from the embedding $v_i$, we employ a pre-trained frozen decoder $\pi$. While the text encoder $f_T$ is initialized from a pre-trained text encoder, one can use any off-the-shelf pre-trained text decoder for $\pi$. Thus, the encoder and the decoder may have different embedding dimensions. To address this potential difference, we introduce a learnable projector $W$ that maps the encoder output to the decoder input. Specifically, given the text embedding $v_i$, we apply a linear projection to obtain $h_i = W^\top v_i$, which is used as the input to the decoder $\pi$. Then, for each $k \in [K]$, the decoder predicts the $k$-th alternative caption $\mathbf{y}_i^{(k)} = [y_{i,1}^{(k)}, \ldots, y_{i,L}^{(k)}]$ composed of $L$ tokens, conditioned on the projected embedding $h_i$. Thus, the token-level reconstruction loss is defined as

$$\mathcal{L}_{\text{Token Reconstruction}} = -\frac{1}{B \cdot K} \sum_{i=1}^{B} \sum_{k=1}^{K} \log \pi \left( \mathbf{y}_i^{(k)} \mid h_i \right), \tag{5}$$

where the log-likelihood value is

$$\log \pi \left( \mathbf{y}_i^{(k)} \mid h_i \right) = \sum_{t=1}^{L} \log \pi \left( y_{i,t}^{(k)} \mid y_{i,<t}^{(k)}, \ h_i \right). \tag{6}$$

Now we discuss how the proposed loss is beneficial for improving the compositional reasoning performance. Since the decoder $\pi$ is frozen and conditioned solely on the text embedding $v_i$, the proposed reconstruction loss trains the text encoder $f_T$ to embed a caption $T_i$ such that a frozen pre-trained decoder can reconstruct the tokens in the alternative caption. The encoder $f_T$ is thus encouraged to capture the relationships between words within a caption, as these are necessary for reconstructing its alternatives, which promotes the compositional reasoning.

### 3.3 Sentence-Level Alignment Loss

Effective compositional reasoning requires not only understanding the relationships between words within a sentence, but also recognizing semantic similarity even when sentences convey the same meaning using different expressions. To this end, as shown in the right-hand side of Fig. 2, we additionally employ a sentence-level alignment loss to explicitly align text embeddings of paraphrased captions that describe the same image. For each image-text pair $(I_i, T_i)$, we generate a paraphrase $T_i'$ of $T_i$ through augmentation, forming a text pair $(T_i, T_i')$. Here, the pair $(T_i, T_i')$ is treated as positive, while paraphrases $(T_i, T_j')$ from other samples in the batch serve as negatives. The sentence-level alignment loss is then defined as

$$\mathcal{L}_{\text{Sentence Alignment}} = -\frac{1}{B} \sum_{i=1}^{B} \log \frac{\phi(T_i, \ T_i')}{\sum_{j=1}^{B} \phi(T_i, \ T_j')}. \tag{7}$$

where $\phi$ is the similarity metric defined in Sec. 3.1 with slight abuse of notation[2]. This alignment loss encourages the encoder to embed paraphrased captions $(T_i, T_i')$ close together in the embedding space, thus letting the encoder capture the semantic relationships between sentences that express the same meaning using different wording and phrasing.

### 3.4 Fine-Tuning Loss of The READ Method

The fine-tuning loss used for the READ method combines the above components:

$$\mathcal{L}_{\text{READ}} = \mathcal{L}_{\text{Contrastive}} + \alpha \, \mathcal{L}_{\text{Token Reconstruction}} + \beta \, \mathcal{L}_{\text{Sentence Alignment}}, \tag{8}$$

where $\alpha$ and $\beta$ are hyperparameters controlling the relative contribution of the auxiliary losses. Together, these losses operate in a complementary way by capturing relational structure at different levels: the token-level reconstruction loss captures relationships between words within a sentence, while the sentence-level alignment loss captures semantic similarity across paraphrased sentences.

---

[2]The definition of $\phi$ in Sec. 3.1 compares an image and a sentence, while here we compare sentences

Table 1: Compositional reasoning performance (%) of the pre-trained CLIP model (ViT-B/32, top row) and its fine-tuned variants (rows 2–7) across five major benchmarks. All models are fine-tuned on 100K samples from the MS-COCO dataset [35]. Among various fine-tuning methods, READ-CLIP achieves the highest average accuracy of 64.1%.

| Models | WhatsUp | VALSE | CREPE | SugarCrepe | SugarCrepe++ | | Avg. |
| | | | | | ITT | TOT | |
| --- | --- | --- | --- | --- | --- | --- | --- |
| CLIP [45] (ViT-B/32) | 41.0 | 67.4 | 23.9 | 73.2 | 60.0 | 46.7 | 52.0 |
| *Fine-tuned: MS-COCO, 100K Samples* | | | | | | | |
| Triplet-CLIP [41] | 41.6 | 64.2 | 15.0 | 82.7 | 61.7 | 57.4 | 53.8 |
| GNM-CLIP [50] | 41.6 | 70.7 | 17.4 | 77.9 | 60.2 | 60.0 | 54.6 |
| CE-CLIP [68] | 40.7 | _76.0_ | 34.8 | _86.0_ | 55.7 | 57.0 | 58.4 |
| NegCLIP [64] | _42.4_ | 73.7 | 30.5 | 83.6 | 65.0 | 62.5 | 59.6 |
| FSC-CLIP [38] | 39.8 | 74.4 | **42.5** | 85.2 | _67.9_ | _64.4_ | 62.4 |
| **READ-CLIP (Ours)** | **43.9** | **76.2** | _41.5_ | **87.0** | **69.8** | **66.2** | **64.1** |

# 4 Experiments

In this section, we empirically evaluate the effectiveness of our proposed READ method. To be specific, we fine-tune the pre-trained CLIP model using the READ method, where the fine-tuned model is dubbed as READ-CLIP. We compare READ-CLIP with various baselines on major compositional reasoning benchmarks. We begin in Sec. 4.1 by describing the experimental setup and present the experimental results in Sec. 4.2. Codes are available at this GitHub repository.

## 4.1 Experimental Setup

**Training:** We use the MS-COCO dataset [35] for all experiments. We follow training practices established in prior work on compositional reasoning [38, 54, 64, 68], using a 100K subsample with the Karpathy split [25], 5 training epochs, a batch size of 256, and the ViT-B/32 architecture. As defined in Eq. 8, our training loss consists of three components: the standard contrastive loss, the token-level reconstruction loss, and the sentence-level alignment loss. We provide implementation details for each component, including hyperparameters, and other specifics, in Appendix A.1.

**Baselines:** We compare READ-CLIP against recent state-of-the-art fine-tuning methods designed to improve compositional reasoning in VLMs. To evaluate against a method relying solely on hard negatives, we include NegCLIP [64], which uses rule-based negatives. To compare with methods that construct synthetic image-text negative pairs, we include GNM-CLIP [50] and Triplet-CLIP [41]. We also consider methods that improve the effectiveness of hard negative captions by incorporating multiple contrastive objectives, including CE-CLIP [68] and FSC-CLIP [38].

**Evaluation:** We evaluate READ-CLIP and the baselines on five benchmarks—WhatsUp [24], CREPE [37], VALSE [40], SugarCrepe [19], and SugarCrepe++ [11]—each designed to assess a different aspect of compositional reasoning. All benchmarks are evaluated using accuracy, which measures whether positive pairs are ranked above all negatives. For each benchmark containing multiple subtasks, we report the accuracy averaged over the subtasks, following prior work [38, 68]. Details of each benchmark are provided in Appendix A.2.

## 4.2 Results

**READ-CLIP outperforms baselines on various compositional benchmarks.** Table 1 reports compositional reasoning performance of READ-CLIP and baselines across five benchmarks. READ-CLIP achieves the highest average accuracy of 64.1%, outperforming the pre-trained CLIP by 12.1%, NegCLIP—a strong and widely cited baseline—by 4.5%, and the second-best model FSC-CLIP by 1.7%. Notably, READ-CLIP ranks first on four benchmarks and second on the remaining one, demonstrating consistently strong performance. This consistent outperformance highlights the advantage of enhancing compositional reasoning in the text encoder through the READ method.

**Reconstruction and alignment losses provide complementary benefits.** We conduct an ablation study to analyze the benefits of the two auxiliary losses introduced in the READ method: token-level reconstruction and sentence-level alignment, summarized in Table 2. Compared to the contrastive-only baseline in row 1, adding token-level reconstruction in row 2 improves average accuracy from

Table 2: Ablation study analyzing how the reconstruction and alignment objectives in our proposed READ method contribute to its performance of READ-CLIP, both individually and jointly. The reconstruction loss improves accuracy on WhatsUp, CREPE, VALSE, and SugarCrepe. The inclusion of the alignment loss improves SugarCrepe++. The combination of both losses results in the highest overall accuracy, indicating their combined contributions to compositional reasoning.

| | Reconstruction Loss | Alignment Loss | WhatsUp | VALSE | CREPE | SugarCrepe | SugarCrepe++ ITT | SugarCrepe++ TOT | Avg. |
|---|---|---|---|---|---|---|---|---|---|
| (1) | | | 40.5 | 74.7 | 38.4 | 86.4 | 69.3 | 66.0 | 62.2 |
| (2) | ✓ | | 43.6 | **76.6** | **41.6** | 86.9 | 69.7 | 64.8 | 63.9 |
| (3) | | ✓ | 43.0 | 75.5 | 40.6 | 86.8 | **70.2** | **67.0** | 63.8 |
| (4) | ✓ | ✓ | **43.9** | 76.2 | 41.5 | **87.0** | 69.8 | 66.2 | **64.1** |

Table 3: Analysis of the impact of key hyperparameter selection on the average accuracy (%) of our proposed READ method: weights for the token reconstruction loss ($\alpha$) and sentence alignment loss ($\beta$), the number of target sequences ($K$) used in the token reconstruction loss, and the size of the T5 [46] decoder model employed for computing the reconstruction loss. Gray cells indicate the hyperparameter configuration that yields the highest average accuracy.

| $\alpha$ (Token Reconst. Loss) | | $\beta$ (Sentence Align. Loss) | | $K$ (Num. of Targets) | | Decoder Model (T5 [46]) | |
|---|---|---|---|---|---|---|---|
| Value | Avg. Acc. | Value | Avg. Acc. | Value | Avg. Acc. | Size | Avg. Acc. |
| - | 63.8 | - | 63.9 | - | 63.8 | - | 63.8 |
| 0.1 | 64.1 | 0.1 | 63.4 | 1 | 64.1 | Small | 64.0 |
| 0.2 | 64.0 | 0.2 | 64.0 | 2 | 63.5 | Base | 64.1 |
| 0.5 | 63.4 | 0.5 | 64.1 | 3 | 63.9 | Large | 64.1 |
| 1.0 | 62.7 | 1.0 | 64.0 | 4 | 64.0 | XL | 63.4 |
| 2.0 | 61.8 | 2.0 | 63.8 | 5 | 63.7 | XXL | 63.9 |

62.2% to 63.9%, notably enhancing WhatsUp by 3.1%, VALSE by 1.9%, and CREPE by 3.2%, while adding sentence-level alignment in row 3 substantially enhances SugarCrepe++ ITT to 70.2% and TOT to 67.0%. Combining both losses in row 4 achieves the highest average accuracy of 64.1%, confirming that each objective offers complementary benefit.

**READ-CLIP is robust to hyperparameter selection.** To assess the robustness of READ, we analyze its sensitivity to four major hyperparameters used for training READ-CLIP: the weight $\alpha$ and $\beta$ for the auxiliary objectives in the READ method, the number $K$ of target sequences, and the size of the T5 decoder [46] used for reconstruction. Table 3 summarizes the average accuracy across the five benchmarks for each configuration. The performance of READ-CLIP remains stable over a wide range of hyperparameter values. In particular, varying the loss weights $\alpha$ and $\beta$ results in only modest performance differences. Also, it achieves strong performance even with a single target sequence ($K = 1$) and a T5-Large decoder, demonstrating robust gains with minimal computational overhead.

**READ provides consistent gains when applied to diverse fine-tuning methods.** The experiments so far applied READ to a modified CLIP objective (Eq. 4) that incorporates hard negative captions only. To establish broader applicability, it is crucial to verify whether the READ method consistently provides gains when applied on top of diverse fine-tuning methods. Therefore, we evaluate READ alongside three baselines: (1) naive CLIP with standard contrastive loss (Eq. 1); (2) NegCLIP [41], and (3) FSC-CLIP [38], without additional hyperparameter tuning. Table 4 presents the results of applying the READ to these three fine-tuning baselines, illustrating its impact across diverse fine-tuning methods. Across all three settings, augmenting the baseline with READ leads to consistent performance improvements on the majority of benchmarks. When applied to naive CLIP, READ improves the average accuracy from 57.0% to 58.2%, with consistent gains across all five benchmarks. For NegCLIP and FSC-CLIP, the average accuracy increases from 61.5% to 62.5% and from 62.8% to 63.6%, respectively. In both cases, READ leads to clear improvements on WhatsUp, VALSE, CREPE, and SugarCrepe, while maintaining comparable performance on SugarCrepe++.

Table 4: Performance comparison of baseline fine-tuning methods—CLIP [45], NegCLIP [64], and FSC-CLIP [38]—and their READ-augmented counterparts, to assess the effectiveness of READ method. READ consistently improves performance on WhatsUp, VALSE, CREPE, and SugarCrepe, while preserving comparable accuracy on SugarCrepe++.

| Methods | WhatsUp | VALSE | CREPE | SugarCrepe | SugarCrepe++ | | Avg. |
|---|---|---|---|---|---|---|---|
| | | | | | ITT | TOT | |
| *Fine-tuned: MS-COCO, 100K Samples, 5 epoch* | | | | | | | |
| CLIP | 41.3 | 70.4 | 15.5 | 81.8 | 66.2 | 66.5 | 57.0 |
| + *READ* | 43.3 (+2.0) | 71.3 (+0.9) | 17.3 (+1.8) | 83.0 (+1.2) | 68.2 (+2.0) | 66.2 (-0.3) | 58.2 (+1.2) |
| NegCLIP | 41.3 | 75.4 | 34.4 | 84.5 | 68.0 | 65.6 | 61.5 |
| + *READ* | 43.7 (+2.4) | 76.5 (+1.1) | 36.7 (+2.3) | 85.2 (+0.7) | 68.1 (+0.1) | 64.9 (-0.7) | 62.5 (+1.0) |
| FSC-CLIP | 41.3 | 73.9 | 42.7 | 85.8 | 68.1 | 65.1 | 62.8 |
| + *READ* | 43.2 (+1.9) | 74.4 (+0.5) | 45.1 (+2.4) | 86.6 (+0.8) | 67.1 (-1.0) | 64.8 (-0.3) | 63.6 (+0.8) |

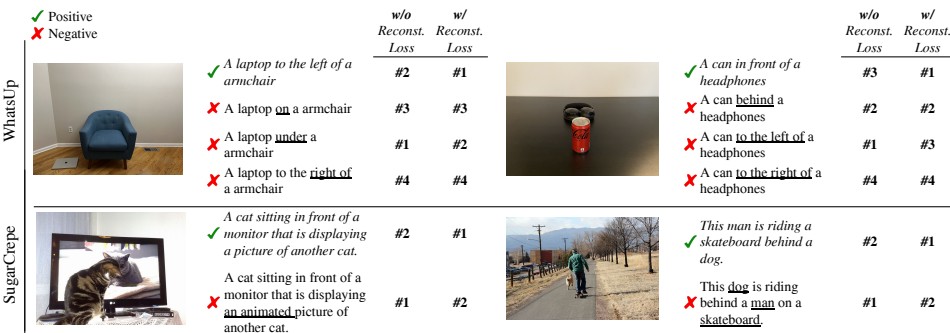

Figure 3: Representative examples illustrating how applying the *reconstruction loss* affects caption rankings based on image-caption cosine similarity on WhatsUp [24] and SugarCrepe [19]. Positive (✓) and negative captions (✗) are shown with their rankings based on image-caption cosine similarity.

## 5 Analysis

Our results in Sec. 4 show that the reconstruction and alignment losses in the READ method consistently improve compositional reasoning, with each offering complementary benefits. To better understand the role of our proposed loss, we present a qualitative analysis in Sec. 5.1 and Sec. 5.2. We then quantitatively analyze the key components of each loss. In Sec. 5.3, we examine the benefit of reconstructing an alternative caption instead of the original one within the *token-level reconstruction* loss. Subsequently, Sec. 5.4 investigates how the quality and diversity of LLM-generated paraphrases used in the *sentence-level alignment* loss affect compositional reasoning.

### 5.1 Token-Level Reconstruction Enhances Encoding of Compositional Relationships

Fig. 3 illustrates the effect observed in Table 2 (rows 1 vs. 2), showing that incorporating the reconstruction loss improves compositional reasoning across benchmarks. Specifically, we observe that the reconstruction loss helps lower the ranking of negative captions that differ from positive ones. These negative captions typically involve subtle structural edits—such as swapping, replacing, or inserting single words or short phrases—that preserve most of the original wording while altering the underlying meaning. This improved discrimination between correct captions and their negative counterparts suggests that the reconstruction loss enables the encoder to recognize semantic differences between those captions by capturing the relationships between words.

### 5.2 Sentence-Level Alignment Promotes Semantic Consistency

Fig. 4 illustrates the effect observed in Table 2 (rows 1 vs. 3), showing how the alignment loss enhances compositional reasoning by affecting the ranking of two positive captions in SugarCrepe++ [11],

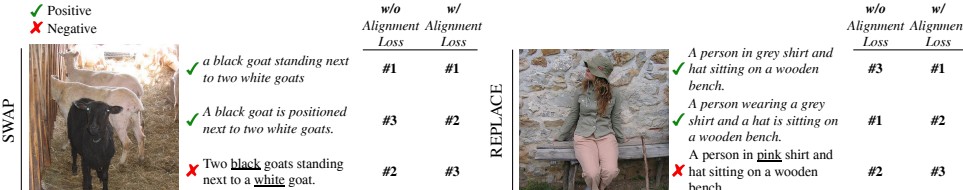

Figure 4: Representative examples from each category (SWAP and REPLACE) of SugarCrepe++ [11] dataset, showing how applying the *alignment loss* improves the ranking of positive captions. In Sugarcrepe++, each image is paired with two positive captions (✓) that are worded differently. In ITT (image-to-text) evaluation, a prediction is considered accurate if both positive captions are ranked higher than all negatives based on image-caption cosine similarity.

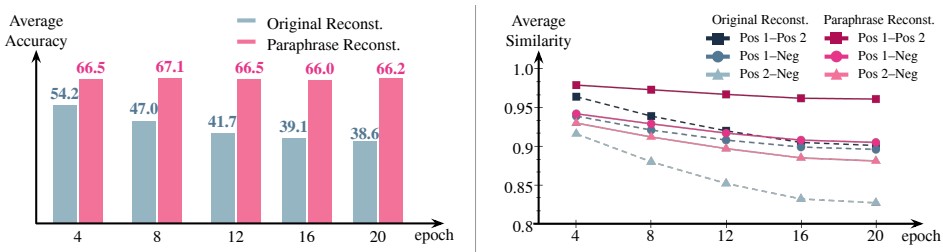

Figure 5: Comparison of reconstructing a paraphrased caption versus the original one, measured by the performance of trained encoder on SugarCrepe++ [11] TOT (text-to-text) benchmark, for various training epochs. In the TOT evaluation, average accuracy (**left**) is computed by checking whether the cosine similarity between the positive caption pair is higher than that of any positive–negative pairs. Average similarity (**right**) measures cosine similarity between caption pairs. Here, `Pos1` and `Pos2` are positive pairs, while others are negative pairs.

where each image is paired with two paraphrases (denoted as `Pos1` and `Pos2`) that convey the same meaning but differ in expression. We observe that applying the alignment loss leads to improved ranking for some positive captions that were previously ranked below negatives. This effect arises because the alignment loss encourages the encoder to embed paraphrased captions closer together in the embedding space, despite differences in wording. Consequently, both captions become embedded closer to the corresponding image than to negative caption, which in turn strengthens the ability in vision-language compositional reasoning.

## 5.3 Reconstructing Alternative Captions Mitigates Overfitting and Enhances Generalization

Recall that the reconstruction loss in Sec. 3.2 is designed to reconstruct an *alternative* caption rather than the *original* one. In here, we explore the effect of using such alternative caption on the compositional reasoning performance of the trained encoder. To be specific, we conduct experiments on the SugarCrepe++ TOT (text-to-text) benchmark, comparing two variants of READ-CLIP: one that reconstructs the original caption and the other that reconstructs an alternative caption.

The left plot in Fig. 5 shows that reconstructing an alternative caption leads to significantly more stable accuracy across training epochs, whereas using the original caption results in a gradual performance decline. To better understand this effect, in the right plot of Fig. 5, we compare the similarity of positive/negative caption pairs in the trained embedding space. We compare three pairs, where `Pos1-Pos2` indicates the positive pair, and other pairs are negative. One can confirm that reconstructing the original caption causes a steady decline in similarity between `Pos1` and `Pos2` over time, which is undesired. This phenomenon can be interpreted as follows: when trained to exactly reconstruct the *original* caption, the encoder increasingly overfits to exact wording and phrasing rather than capturing the underlying relationships between words within a caption. In contrast, reconstructing an *alternative* caption better preserves semantic similarity between positive captions and maintains greater discrimination from negatives. These findings indicate that reconstructing an alternative caption mitigates overfitting to exact wording and enhances generalization in relational understanding.

Table 5: Compositional reasoning performance under varying *quality* of LLM-generated paraphrases in Eq. 7. We randomly replaced 10% or 20% of LLM-generated paraphrases with unrelated captions from the dataset to simulate lower-quality paraphrases during training.

| Model | WhatsUp | VALSE | CREPE | SugarCrepe | SugarCrepe++ | | Avg. |
|---|---|---|---|---|---|---|---|
| | | | | | ITT | TOT | |
| READ-CLIP | 43.9 | 76.2 | 41.5 | 87.0 | 69.8 | 66.2 | 64.1 |
| READ-CLIP (10% Noise) | 43.6 | 76.0 | 39.0 | 86.9 | 67.1 | 64.7 | 62.9 |
| READ-CLIP (20% Noise) | 43.4 | 76.0 | 38.6 | 86.8 | 65.1 | 62.9 | 62.1 |

Table 6: Compositional reasoning performance under varying *diversity* of LLM-generated paraphrases in Eq. 7. We generated multiple paraphrases per caption and randomly sampled one during each training step to examine whether increased diversity improves robustness.

| Model | WhatsUp | VALSE | CREPE | SugarCrepe | SugarCrepe++ | | Avg. |
|---|---|---|---|---|---|---|---|
| | | | | | ITT | TOT | |
| READ-CLIP ($\text{num}_p = 1$) | 43.9 | 76.2 | 41.5 | 87.0 | 69.8 | 66.2 | 64.1 |
| READ-CLIP ($\text{num}_p = 3$) | 43.4 | 76.4 | 41.1 | 86.5 | 70.0 | 66.4 | 64.0 |
| READ-CLIP ($\text{num}_p = 5$) | 43.6 | 76.0 | 41.3 | 86.5 | 70.8 | 66.6 | 64.1 |

## 5.4 Sentence-Level Alignment is Robust to Quality and Diversity of Paraphrases

Recall that the sentence-level alignment loss in Sec. 3.3 uses LLM-generated paraphrases to encourage semantic consistency between captions with different wordings. In here, we investigate how the *quality* and *diversity* of these paraphrases affect compositional reasoning performance.

First, we assess whether the performance of compositional reasoning of our proposed READ method is sensitive to lower-quality paraphrases. To this end, we intentionally inject noise by randomly replacing LLM-generated paraphrases with unrelated captions from the dataset at two levels: 10% and 20%. As shown in Table 5, performance drops by only 1.2–2.0% on average when 10–20% of paraphrases are replaced with noise, indicating that the sentence-level alignment loss is reasonably robust to moderate degradation in paraphrase quality.

Next, we examine whether increasing diversity of LLM-generated paraphrases improves the performance of compositional reasoning. We vary the number of LLM-generated paraphrases per caption ($\text{num}_p \in \{1, 3, 5\}$) and randomly sample one at each training step. All other components in Eq. 7 remain unchanged, thereby the only difference is the diversity of available paraphrases. Table 6 shows that while increasing $\text{num}_p$ slightly improves performance on SugarCrepe++ [11] – where recognizing paraphrased captions as semantically equivalent is critical – the average performance across all benchmarks remains nearly unchanged. This suggests that a single number of LLM-generated paraphrase is already sufficient for effective alignment, and additional diversity provides only marginal benefits.

## 6 Conclusion

We introduced READ, a fine-tuning method that enhances compositional reasoning in contrastively trained VLMs by integrating token-level reconstruction and sentence-level alignment objectives. READ explicitly captures compositional relationships, enabling READ-CLIP to outperform other fine-tuning baselines across diverse benchmarks. We hope this work provides a practical approach for compositionality-aware fine-tuning of VLMs, and encourages further exploration of auxiliary objectives to strengthen the compositional reasoning ability of text encoders.

**Limitation.** While our method is designed to leverage multiple captions per image, it can still be applied to the dataset with only a single caption per image by generating additional paraphrases using LLMs, although this introduces additional complexity. In addition, we only used T5 [46] decoder in our reconstruction loss, without exploring the impact of alternative generative architectures [16, 44]. We did not assess the effect of fine-tuning the decoder as well, which may influence the compositional reasoning capability of our proposed method.

## Acknowledgements

This work was partially supported by the National Research Foundation of Korea (NRF) grant funded by the Ministry of Science and ICT (MSIT) of the Korean government (RS-2024-00345351, RS-2024-00408003), and Institute of Information & Communications Technology Planning & Evaluation (IITP) grant funded by MSIT (RS-2023-00259934, RS-2025-02283048).

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

Figure 6: A prompt for generating synthetic paraphrased captions.

| Benchmark | License | Image Source |
|---|---|---|
| CREPE-Productivity [37] | Unspecified | Visual Genome [27] |
| SugarCrepe [19] | MIT | COCO [35] |
| SugarCrepe++ [11] | MIT | COCO [35] |
| WhatsUp [24] | MIT | Custom-collected, COCO [35], GQA [21] |
| VALSE [40] | MIT | Visual7W [73], COCO [35], SWiG [43], FOIL-it [52] |

Table 7: Benchmarks used in the evaluation, along with license and image source.

# A  Appendix

## A.1  Implementation Details

**Batch Sampling for Training.** As described in Sec. 4.1, all our experiments are conducted using the COCO [35] dataset. The COCO dataset is an image-caption dataset consisting of images, each associated with a set of captions. During training, we first sample a batch of $B$ images from this dataset. For each $i \in \{1, 2, \ldots, B\}$, let $\mathcal{T}_i = \{T_i^{(n)}\}_{n=1}^N$ denote the set of $N$ captions associated with $I_i$ (where $I_i$ denotes the $i$-th image). From this set, we randomly select one caption $T_i \in \mathcal{T}_i$ to form a positive image-text pair $(I_i, T_i)$. In this way, we obtain a batch of $B$ image-text pairs, denoted as $\{(I_i, T_i)\}_{i=1}^B$.

**Details of Each Loss Component.** Recall that our proposed *REconstruction and Alignment of text Descriptions* (READ) method comprises three components in the final loss (Eq. 8): the standard *contrastive* loss, the token-level *reconstruction* loss, and the sentence-level *alignment* loss. For the *contrastive* loss (Eq. 4), we incorporate $M = 3$ hard negative captions per image, generated via rule-based perturbations as proposed in NegCLIP [64]. For the *reconstruction* loss (Eq. 5), we use a frozen decoder extracted from the pre-trained encoder-decoder language model, `T5-Large` [46]. To obtain $\{y_i^{(k)}\}_{k=1}^K$, we randomly sample $K = 1$ element from the caption set $\mathcal{T}_i$ associated with each image-text pair $(I_i, T_i)$ and use it as an alternative caption. For the *alignment* loss (Eq. 7), we generate a paraphrased caption $T_i'$ for each $T_i$ via augmentation using large language models. Specifically, prior to training, we generate one paraphrased caption for every caption in each image's original caption set using the `gpt-4o-mini-2024-07-18` model [22]. This is done by applying a simple prompt as shown in Fig.6, with a temperature of 1.0 and all other parameters set to their default values [22]. From this augmentation, we obtain a synthetic caption set for each image. Given a batch of sampled image-text pairs $\{(I_i, T_i)\}_{i=1}^B$ during training, we randomly sample one caption from the union of the original and synthetic caption sets associated to $I_i$, and use it as $T_i'$. Finally, the weighting factors in Eq. 8 are set to $\alpha = 0.1$ and $\beta = 0.5$.

**Training Details.** We fine-tune all models using the Huggingface `transformers` [61] library[3]. The AdamW optimizer is used with a learning rate of $1.0 \times 10^{-5}$, cosine annealing schedule, 50 warmup steps, and a weight decay of 0.1. Training is performed with `bf16` mixed precision for computational efficiency. All experiments are conducted using a single A100 40GB GPU.

## A.2  Evaluation Details

**Description of Benchmarks. WhatsUp** [24] evaluates spatial reasoning by testing whether models can interpret relative object positions. **CREPE** [37] measures compositional reasoning at varying

---

[3] https://github.com/huggingface/transformers

Table 8: Detailed results on CREPE [37].

| Model | Atom | Negate | Swap | Total |
|---|---|---|---|---|
| CLIP [45] (Pre-trained) | 18.9 | 35.3 | 17.3 | 23.9 |
| Triplet-CLIP [41] | 18.5 | 11.2 | 15.3 | 15.0 |
| GNM-CLIP [50] | 21.7 | 13.3 | 17.3 | 17.4 |
| CE-CLIP [68] | 40.4 | 21.2 | 42.7 | 34.8 |
| NegCLIP [64] | 32.1 | 16.6 | 42.8 | 30.5 |
| FSC-CLIP [38] | **40.5** | **41.3** | 45.5 | **42.5** |
| **READ-CLIP** | 37.1 | 26.2 | **61.2** | 41.5 |

Table 9: Detailed results on WHATSUP [24].

| Model | COCO-Spatial | GQA-Spatial | Whats-up | Total |
|---|---|---|---|---|
| CLIP [45] (Pre-trained) | 44.5 | 47.8 | 30.7 | 41.0 |
| Triplet-CLIP [41] | 49.2 | 47.1 | 28.5 | 41.6 |
| GNM-CLIP [50] | 44.8 | 47.4 | 32.6 | 41.6 |
| CE-CLIP [68] | 43.7 | 47.8 | 30.7 | 40.7 |
| NegCLIP [64] | 45.1 | 47.7 | 34.4 | 42.4 |
| FSC-CLIP [38] | 47.7 | 41.9 | 29.6 | 39.8 |
| **READ-CLIP** | **51.6** | **48.1** | **31.8** | **43.9** |

complexity levels using logical operations such as conjunction, negation, and attribute swapping. **VALSE** [40] assesses fine-grained linguistic understanding, including object existence, quantity, action semantics, and coreference resolution. **SugarCrepe** [19] focuses on relational reasoning through hard negative captions crafted with natural linguistic variation. **SugarCrepe++** [11] extends SugarCrepe by adding a paraphrased positive caption and introduces two tasks: (1) image-to-text (ITT), which tests whether all paraphrased positives for a given image are ranked above all negatives, and (2) text-to-text (TOT), which evaluates semantic consistency by checking whether each positive paraphrase pair is ranked above all negative pairs in the absence of visual context. Since our study aims to improve the compositional reasoning capability of VLMs such as CLIP, we primarily adopt the ITT metric as a major focus for evaluation, while including TOT as a supplementary measure.

**Licensing of the Benchmarks.** We conduct our evaluation on five publicly available compositional reasoning benchmarks. Table 7 summarizes their license information and image sources. All datasets used for training and evaluation are either MIT-licensed or publicly released for research use.

### A.3 Supplementary Experimental Results

This section provides extended experimental results that complement the main paper. We include both (1) detailed, category-wise results for each benchmark and (2) additional evaluations on zero-shot image classification datasets [5] to further assess the generalization ability of READ-CLIP.

### A.3.1 Detailed Version of Experimental Results in the Main Paper

To complement the results presented in Table 1, we report category-wise results on all evaluation benchmarks, including CREPE, VALSE, WhatsUp, SugarCrepe, and SugarCrepe++ (ITT and TOT). Table 8–13 present detailed breakdowns for individual benchmarks. These results provide a finer-grained analysis of compositional reasoning performance supplementing the aggregated scores shown in the main paper. Overall, the detailed results confirm that READ-CLIP consistently improves over baselines across categories. In addition, we provide Fig. 7 and Fig. 8, which present extended qualitative examples that respectively complement Fig. 3 and Fig. 4 in the main paper.

Table 10: Detailed results on VALSE [40].

| Model | Actions | Coreference | Counting | Existence | Noun Phrases | Plurals | Relations | Total |
|-------|---------|-------------|----------|-----------|--------------|---------|-----------|-------|
| CLIP [45] (Pre-trained) | 74.3 | 54.4 | 61.7 | 69.3 | 90.4 | 57.9 | 66.0 | 67.4 |
| Triplet-CLIP [41] | 72.6 | 54.8 | 54.0 | 59.4 | 91.4 | 61.3 | 60.9 | 64.2 |
| GNM-CLIP [50] | 72.1 | **61.1** | 66.5 | 76.0 | 90.8 | 68.7 | 62.6 | 70.7 |
| CE-CLIP [68] | 85.0 | 59.9 | 67.6 | 78.2 | 94.4 | **78.8** | 74.0 | 76.0 |
| NegCLIP [64] | 84.1 | 60.2 | 65.7 | 75.3 | 93.4 | 70.3 | 69.6 | 73.7 |
| FSC-CLIP [38] | 82.9 | 59.4 | 66.3 | 77.6 | 93.5 | 72.7 | **75.3** | 74.4 |
| **READ-CLIP** | **86.3** | 55.7 | **69.0** | **80.8** | **95.8** | 73.8 | 75.0 | **76.2** |

Table 11: Detailed results on SugarCrepe [19]. `Att.`, `Obj.`, and `Rel.` denote the targets of transformation: `Attribute`, `Object`, and `Relation`, respectively.

| Model | Add Att. | Add Obj. | Replace Att. | Replace Obj. | Replace Rel. | Swap Att. | Swap Obj. | Total |
|-------|----------|----------|--------------|--------------|--------------|-----------|-----------|-------|
| CLIP [45] (Pre-trained) | 69.5 | 77.0 | 80.3 | 90.7 | 69.4 | 64.1 | 61.2 | 73.2 |
| Triplet-CLIP [41] | 85.5 | 87.5 | 86.7 | 94.5 | **83.2** | 73.1 | 68.6 | 82.7 |
| GNM-CLIP [50] | 79.9 | 88.4 | 84.9 | 93.2 | 67.8 | 70.0 | 61.2 | 77.9 |
| CE-CLIP [68] | **91.9** | **92.3** | 90.2 | 94.4 | 81.4 | 76.7 | 75.1 | 86.0 |
| NegCLIP [64] | 85.3 | 90.0 | 88.2 | 94.0 | 74.6 | 77.9 | 75.5 | 83.6 |
| FSC-CLIP [38] | 86.7 | 90.2 | 89.2 | 94.3 | 80.4 | 77.8 | 77.6 | 85.2 |
| **READ-CLIP** | 87.7 | 90.3 | **91.0** | **94.9** | 80.6 | **82.7** | **81.6** | **87.0** |

Table 12: Detailed results on *image-to-text* subset of SugarCrepe++ [11]. `Att.`, `Obj.`, and `Rel.` denote the targets of transformation: `Attribute`, `Object`, and `Relation`, respectively.

| Model | Replace Att. | Replace Obj. | Replace Rel. | Swap Att. | Swap Obj. | Total |
|-------|--------------|--------------|--------------|-----------|-----------|-------|
| CLIP [45] (Pre-trained) | 65.7 | 87.0 | 56.5 | 45.0 | 45.8 | 60.0 |
| Triplet-CLIP [41] | 71.7 | 87.0 | **62.3** | 48.5 | 39.2 | 61.7 |
| GNM-CLIP [50] | 68.9 | 89.5 | 52.8 | 48.6 | 41.2 | 60.2 |
| CE-CLIP [68] | 62.4 | 81.9 | 53.5 | 40.5 | 40.0 | 55.7 |
| NegCLIP [64] | 69.7 | 89.8 | 52.6 | 58.1 | 54.7 | 65.0 |
| FSC-CLIP [38] | 73.5 | **90.4** | 60.1 | 60.4 | 55.1 | 67.9 |
| **READ-CLIP** | **72.2** | 90.1 | 57.5 | **66.2** | **62.9** | **69.8** |

Table 13: Detailed results on *text-to-text* subset of SugarCrepe++ [11]. `Att.`, `Obj.`, and `Rel.` denote the targets of transformation: `Attribute`, `Object`, and `Relation`, respectively.

| Model | Replace Att. | Replace Obj. | Replace Rel. | Swap Att. | Swap Obj. | Total |
|-------|--------------|--------------|--------------|-----------|-----------|-------|
| CLIP [45] (Pre-trained) | 59.3 | 83.7 | 38.6 | 32.7 | 19.2 | 46.7 |
| Triplet-CLIP [41] | 74.1 | 92.3 | 52.3 | 43.2 | 24.9 | 57.4 |
| GNM-CLIP [50] | 76.9 | 95.9 | 51.9 | 48.9 | 26.1 | 60.0 |
| CE-CLIP [68] | 74.2 | 89.6 | 52.0 | 42.8 | 26.5 | 57.0 |
| NegCLIP [64] | 76.4 | 94.6 | 51.7 | 56.6 | 33.1 | 62.5 |
| FSC-CLIP [38] | **83.5** | 96.3 | 56.8 | 56.3 | 29.0 | 64.4 |
| READ-CLIP | 77.3 | **97.6** | **58.0** | 56.8 | **41.2** | **66.2** |

### A.3.2 Additional Experimental Results

To further verify the generalization capability of READ-CLIP, we conducted an additional evaluation beyond the main benchmarks. We assessed the model's zero-shot image classification performance across 23 widely used benchmarks [5]. We compared READ-CLIP with the original CLIP [45] pre-trained model and six representative compositional reasoning fine-tuning methods.

In Table 14, The results show that all fine-tuned models, including READ-CLIP, achieve lower average performance than the original CLIP across the 23 datasets. This finding aligns with trends reported in previous study [38], where improvements in compositional understanding often come at the cost of general zero-shot capability. Such trade-offs highlight the inherent difficulty of maintaining broad generalization while adapting models specifically for compositional reasoning.

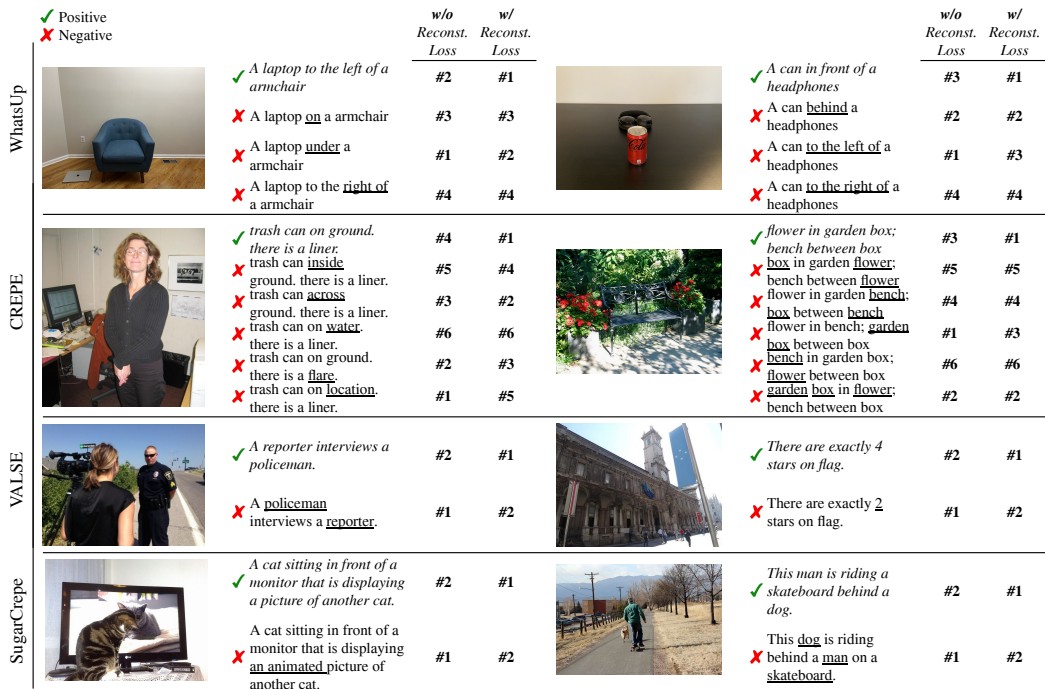

Figure 7: Extended representative examples for Fig. 3, including additional examples from CREPE [37] and VALSE [40], as well as WhatsUp [24] and SugarCrepe [19]. These extended examples additionally include a broader range of benchmarks where applying the *reconstruction loss* proved effective.

Table 14: Additional results of zero-shot image classification performance across various datasets. Here, CLIP [45] is the pre-trained model, while the other models are fine-tuned versions of CLIP [45].

| Model | caltech101 | cars | cifar10 | cifar100 | country211 | dtd | eurosat | fer2013 | fgvc-aircraft-2013b | flowers | food101 | gtsrb | imagenet-o | imagenet-1k | imagenet-sketch | imagenet-v2 | kitti-distance | mnist | pcam | rendered-sst2 | resisc45-clip | stl10 | voc2007classification | Avg. |
|---|---|---|---|---|---|---|---|---|---|---|---|---|---|---|---|---|---|---|---|---|---|---|---|---|
| CLIP [45] | 81.5 | **59.7** | 89.8 | 64.3 | **17.2** | **44.3** | 50.5 | 41.2 | **19.7** | **66.4** | **84.0** | **32.5** | **47.6** | **63.4** | **42.3** | **55.7** | 27.1 | 48.3 | **62.3** | 58.8 | 53.6 | **97.1** | 76.5 | **55.8** |
| NegCLIP [64] | **82.6** | 53.9 | 88.9 | 63.0 | 15.0 | 43.0 | 49.7 | 46.7 | 16.8 | 65.0 | 79.4 | 30.2 | 46.5 | 60.9 | 40.4 | 53.2 | 27.7 | 49.7 | 54.9 | 58.6 | 52.9 | 96.7 | **79.6** | 54.6 |
| GNM-CLIP [50] | 81.5 | 53.1 | 88.5 | **65.0** | 15.2 | 42.1 | 50.7 | 46.0 | 17.2 | 63.3 | 81.8 | 30.2 | 47.4 | 61.4 | 41.0 | 54.1 | 25.3 | **54.3** | 55.6 | 58.5 | 49.8 | 96.4 | 77.4 | 54.6 |
| FSC-CLIP [38] | 81.8 | 51.8 | 89.1 | 64.9 | 14.5 | 40.7 | 51.6 | 49.5 | 15.8 | 61.7 | 78.7 | 29.8 | 45.5 | 59.2 | 38.9 | 51.7 | 29.4 | 50.4 | 51.0 | **59.8** | 52.8 | 96.1 | 79.0 | 54.1 |
| DAC-LLM [6] | 77.7 | 39.4 | **90.4** | 63.9 | 14.3 | 39.0 | **52.3** | **50.5** | 11.3 | 54.6 | 74.2 | 24.2 | 45.5 | 51.0 | 35.2 | 45.0 | 16.6 | 42.2 | 50.0 | 54.4 | 49.6 | **97.1** | 77.9 | 50.3 |
| DAC-SAM [6] | 75.7 | 39.9 | 89.9 | 63.7 | 14.8 | 40.0 | 51.2 | 47.7 | 9.0 | 53.9 | 72.3 | 24.9 | 45.5 | 52.4 | 35.1 | 46.8 | 18.7 | 45.3 | 50.0 | 59.8 | 51.7 | 96.1 | 65.8 | 50.0 |
| **READ-CLIP** | 78.2 | 39.6 | 87.1 | 57.8 | 10.2 | 35.0 | 39.2 | 41.0 | 13.1 | 52.2 | 71.6 | 26.7 | 44.5 | 51.5 | 32.9 | 45.3 | 30.5 | 48.0 | 47.3 | 52.3 | 44.3 | 95.2 | 78.9 | 48.8 |
| CE-CLIP [68] | 78.3 | 35.3 | 85.9 | 60.1 | 9.5 | 35.2 | 42.8 | 39.5 | 10.0 | 48.2 | 70.1 | 28.0 | 44.8 | 49.9 | 31.5 | 43.2 | **34.6** | 40.6 | 50.0 | 61.2 | 47.7 | 95.8 | 77.3 | 48.7 |
| Triplet-CLIP [41] | 80.6 | 23.9 | 89.1 | 61.5 | 7.1 | 39.3 | 35.2 | 47.7 | 12.7 | 54.6 | 76.3 | 24.7 | 42.8 | 54.8 | 37.0 | 48.4 | 15.3 | 34.3 | 49.6 | 51.8 | **54.7** | 94.6 | 72.9 | 48.2 |

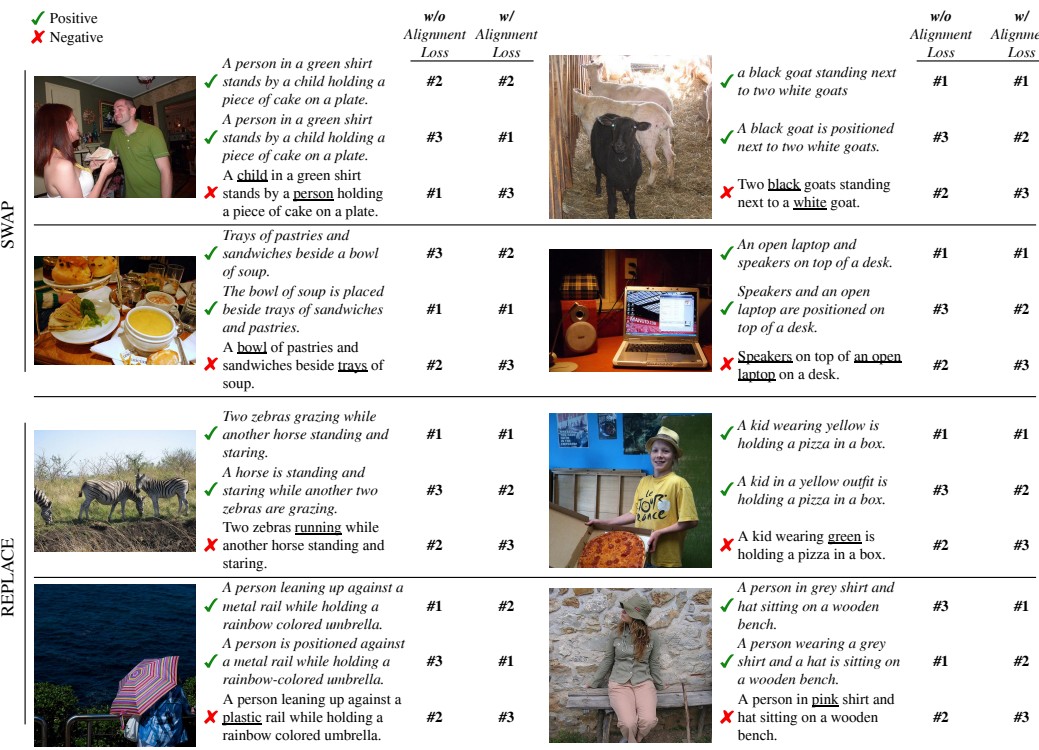

Figure 8: Extended representative examples for Fig. 4, including additional examples each category (SWAP and REPLACE) of SugarCrepe++. These extended examples further illustrate the effectiveness of applying the *alignment loss* across diverse cases.

