# OpenReview forum: "Enhancing Compositional Reasoning in CLIP via Reconstruction and Alignment of Text Descriptions"
_NeurIPS.cc/2025/Conference — NeurIPS 2025 poster_

### Official Review · Reviewer_gRGm · 2025-07-01

**Clarity:** 2
**Significance:** 3
**Originality:** 2
**Rating:** 5
**Confidence:** 3

**Summary:**

This paper introduces READ, a fine-tuning method aimed at enhancing compositional reasoning in VLMs such as CLIP. The authors propose an approach that adds two auxiliary losses to the standard Contrastive Loss:
- A Token-level Reconstruction Loss, which helps the text encoder better capture intra-sentence relationships.
- A Sentence-level Alignment Loss, designed to improve the encoder's ability to recognize semantic similarity between sentences that use different expressions to convey the same meaning.

The authors apply their method to CLIP and its variants, evaluate it on five compositional reasoning benchmarks, and report consistent improvements over existing similar techniques. They also provide ablation studies to assess the contribution of each component of the proposed loss.

**Questions:**

- Clarification on the paraphrasing process: The paper mentions paraphrase generation through augmentation, but the details remain unclear. Did you use a specific LLM to generate paraphrases? If so, which one, and under what settings? Or are you using a VLM to do this, conditioning on both text and image? If a VLM was used for paraphrasing, this could raise a concern about circularity, i.e., relying on VLM, that you criticized for poor compositional reasoning, to improve compositional reasoning of other VLMs. Clarifying this aspect would strengthen the methodological transparency of the paper.
- Highlighting the originality of the contribution: While the reconstruction and alignment losses are motivated and well implemented, their novelty may not be immediately evident to readers who are less familiar with the literature. I recommend emphasizing more explicitly how your approach differs from prior work using reconstruction losses, specifically, by explaining that although similar techniques exist for general-purpose encoder training, they have not been applied to the problem of improving compositional reasoning in contrastive VLMs. Making this distinction more visible could better communicate the originality of your work.
- More qualitative examples: The qualitative examples presented in the main paper are insightful. However, I suggest including additional qualitative examples, especially failure cases and edge scenarios, maybe in the appendix. This would help readers better understand the behavior of the model under different conditions and strengthen the empirical analysis.

**Ethical Concerns:**

["NO or VERY MINOR ethics concerns only"]

**Final Justification:**

The rebuttal effectively addressed my concerns, and the clarifications provided reinforce the validity and relevance of the proposed contributions. I thus maintain my accept rating.

**Limitations:**

yes

**Paper Formatting Concerns:**

See minor weaknesses (w1-5)

**Quality:**

3

**Strengths And Weaknesses:**

*Strenghts*:
- The problem of compositional reasoning is well explained in the introductory sections of the paper, providing useful context and motivation.
- The method achieves strong results, showing a clear overall improvement over other CLIP-based models addressing similar compositional reasoning challenges. In the only benchmark where READ does not rank first, it still ranks second.
- The experimental section is comprehensive and well-structured, covering key aspects of the proposed method. The ablation studies in Table 2 and Table 3 are particularly valuable, and the inclusion of the experiment in Table 4â€”evaluating READ when applied to different fine-tuning baselinesâ€”is especially noteworthy. Overall, Section 4 demonstrates a solid and thorough evaluation of the method.

*Weaknesses*:
- This is more of a minor issue, but some parts of the method section, particularly the mathematical formulation, are difficult to follow. The notation is heavy, with many variables introduced at once, which can make the equations hard to parse on a first read. A clearer presentation of the method, possibly with simplified or modular notation, would improve readability.
- Another potential issue is that, on a first read, the method may appear to require the generation of multiple captions per image, which could be perceived as a limitation. It would be helpful to clarify early in the paper, either in the Introduction or Related Work, that datasets like MS-COCO already exist and provide multiple captions per image, making the method more accessible. Additionally, when MS-COCO is mentioned in Section 4.1, a brief description of its structure would help readers who may not be familiar with the dataset.

---

> ### Author Rebuttal · Authors · 2025-07-31
>
> ## To Reviewer 4 (gRGm)
>
> We thank the Reviewer gRGm for the detailed review and constructive suggestions. We appreciate your acknowledgement that our method achieves strong result, our ablation studies are valuable, and our evaluation is solid and thorough. Please find our answers to your comments and questions as follows.
>
>
> **[R4-1]** `This is more of a minor issue, but some parts of the method section, particularly the mathematical formulation, are difficult to follow. The notation is heavy, with many variables introduced at once, which can make the equations hard to parse on a first read. A clearer presentation of the method, possibly with simplified or modular notation, would improve readability.`
>
> Thank you for highlighting this point. To address this concern, we plan to improve the clarity in the revised manuscript by:
> - **Modularizing the notation** around core conceptual units (e.g., alignment vs. reconstruction),
> - **Adding intuitive summaries** between mathematical blocks to guide the reader through key steps.
>
> We will ensure that these revisions preserve mathematical precision while significantly improving readability. Thank you again for this constructive suggestion.
>
> ---
>
> **[R4-2]** `Another potential issue is that, on a first read, the method may appear to require the generation of multiple captions per image, which could be perceived as a limitation. It would be helpful to clarify early in the paper, either in the Introduction or Related Work, that datasets like MS-COCO already exist and provide multiple captions per image, making the method more accessible. Additionally, when MS-COCO is mentioned in Section 4.1, a brief description of its structure would help readers who may not be familiar with the dataset.`
>
>
> We thank the reviewer for the thoughtful comment.
>
> In the revised manuscript, we will clarify early in the paper (Introduction or Related Work) that datasets such as MS-COCO **already** provide multiple human-written captions per image. This will help readers understand that our method does not require any additional caption generation. Furthermore, when introducing MS-COCO in Section 4.1, we will include a brief description of its structure to better support readers who may be less familiar with the dataset.
>
>
> ---
>
> **[R4-3]** `Clarification on the paraphrasing process: The paper mentions paraphrase generation through augmentation, but the details remain unclear. Did you use a specific LLM to generate paraphrases? If so, which one, and under what settings? Or are you using a VLM to do this, conditioning on both text and image? If a VLM was used for paraphrasing, this could raise a concern about circularity, i.e., relying on VLM, that you criticized for poor compositional reasoning, to improve compositional reasoning of other VLMs. Clarifying this aspect would strengthen the methodological transparency of the paper.`
>
>
>
> Thank you for raising this important point.
>
> In our experiments, paraphrasing was performed using a **text-only LLM**, not a vision-language model. As mentioned in *supplementary material*, to generate paraphrased captions for our sentence-level alignment loss, we use the `gpt-4o-mini-2024-07-18`, and details are as follows:
>
> - Each original caption was provided **as text-only input**.
> - No image or visual features were used during generation.
> - We used a simple prompt (shown in Appendix Fig. 6), with **temperature = 1.0** and all other decoding parameters left at default settings.
> - For every caption in the COCO dataset, we generated **one paraphrased variant** offline before training.
>
> We will include these details in the revised manuscript.
>
>
> ---
>
>
> **[R4-4]** `Highlighting the originality of the contribution: While the reconstruction and alignment losses are motivated and well implemented, their novelty may not be immediately evident to readers who are less familiar with the literature. I recommend emphasizing more explicitly how your approach differs from prior work using reconstruction losses, specifically, by explaining that although similar techniques exist for general-purpose encoder training, they have not been applied to the problem of improving compositional reasoning in contrastive VLMs. Making this distinction more visible could better communicate the originality of your work.`
>
> Thank you for this helpful suggestion. As you rightly pointed out, the core novelty of our work lies in introducing token-level reconstruction and sentence-level alignment objectives into the contrastive training of vision-language models (VLMs)—**not for general-purpose encoding, but explicitly to improve compositional reasoning**.
>
> While text reconstruction and text alignment objectives have appeared in prior work (e.g., for general-purpose language model training), to the best of our knowledge, **our work is the first to adapt these textual supervision for improving compositional generalization in VLMs trained with contrastive learning like CLIP**.
>
> We acknowledge that this distinction may not be immediately clear to readers unfamiliar with the relevant literature. Accordingly, we will revise the manuscript to emphasize this point more explicitly in the Introduction and Related Work sections.
>
>
> ---
>
> **[R4-5]** `More qualitative examples: The qualitative examples presented in the main paper are insightful. However, I suggest including additional qualitative examples, especially failure cases and edge scenarios, maybe in the appendix. This would help readers better understand the behavior of the model under different conditions and strengthen the empirical analysis.`
>
> Thank you for this thoughtful suggestion. As per the reviewer's comment, we will augment the appendix in the revised manuscript with additional qualitative examples, including:
>
> - **Cases where both READ-CLIP and other baselines fail**, highlighting shared limitations in compositional reasoning.
> - **Cases where READ-CLIP fails but other baselines succeed**, which can reveal specific weaknesses unique to our method.

---

> ### Comment · Area_Chair_huKn · 2025-08-05
>
> Reviewer gRGm, please engage in the discussion period. I understand that the review period started over a weekend, but we only have a few days remaining in the (now slightly extended) discussion period. The authors have provided a thoughtful response to your review and you are obligated to respond to it. You should share with the authors if they addressed questions or concerns you had, and seek clarification about any questions or concerns that remain. Please post your response as soon as you can so that there is time for the authors to follow up and discussion to progress as needed.

---

> > ### Comment · Reviewer_gRGm · 2025-08-05
> >
> > Dear authors,
> >
> > Thank you for addressing my concerns and for your detailed responses.
> >
> > I appreciate the clarification about the paraphrase generation (R4-3), and I have a further question.
> >
> > Since the method relies on these paraphrases for both the reconstruction and alignment losses, did you evaluate the impact of their quality and diversity? Do you think it would be possible to use more than one paraphrased variant per caption (i.e., K > 1), in order to improve training robustness or mitigate overfitting to a single rephrasing style?

---

> > > ### Author Response · Authors · 2025-08-07
> > >
> > > **[R4-6]**: `Did you evaluate the impact of the quality and diversity of LLM-generated captions (i.e., paraphrased captions)? Do you think it would be possible to use more than one paraphrased variant per caption (i.e., K > 1), in order to improve training robustness or mitigate overfitting to a single rephrasing style?`
> > >
> > > We thank the reviewer for raising the important point.
> > >
> > > First, we would like to gently clarify a potential point of confusion regarding the usage of LLM-generated captions. Specifically, we only use LLM-generated captions for the sentence *alignment* loss, and do not use LLM-generated captions for the token *reconstruction* loss.
> > >
> > > To address your question on the LLM-generated captions (i.e., paraphrased captions), we provide both (1) the experiment we conducted to assess the *quality* of the paraphrased captions, and (2) an ***additional experiment*** we carried out in response to the reviewer’s question to better evaluate their *diversity*.
> > >
> > > ### (1) On the Quality:
> > >
> > > To assess the impact of paraphrase quality on the model performance, we conducted two experiments: (a) one that evaluates the effect of **low-quality paraphrases** via noise injection, and (b) another that investigates whether using **higher-quality LLMs** to generate paraphrases yields better performance.
> > >
> > > #### (a) Effect of Lower-Quality Paraphrases
> > >
> > > As detailed in `[R2-4]`, we evaluated the **robustness of READ-CLIP when exposed to lower-quality paraphrases**. Specifically, we introduced noise into the paraphrases by randomly replacing a portion of paraphrased captions with unrelated captions from the dataset. We experimented with two settings: 10% and 20% replacement.
> > >
> > > | Model | CREPE | SUGARCREPE | VALSE | WHATSUP | SUGARCREPE++ (ITT) | SUGARCREPE++ (TOT) | AVG  |
> > > |-|-|-|-|-|-|-|-|
> > > | READ-CLIP                | 41.5  | 87.0       | 76.2   | 43.9    | 69.8                | 66.2                | 64.1 |
> > > | READ-CLIP (10% Noise)    | 39.0  | 86.9       | 76.0   | 43.6    | 67.1                | 64.7                | 62.9 |
> > > | READ-CLIP (20% Noise)    | 38.6  | 86.8       | 76.0   | 43.4    | 65.1                | 62.9                | 62.1 |
> > >
> > > As expected, injecting noise into paraphrases leads to performance degradation. However, the drop is relatively modest—**only 1.2 to 2.0 points in average score**—indicating that **READ-CLIP is reasonably robust to moderate paraphrase noise**.
> > >
> > > #### (b) Effect of Higher-Quality Paraphrases
> > >
> > > Additionally, as detailed in `[R3-3]`, we analyzed whether stronger LLMs—**capable of generating higher-quality paraphrases—would improve READ-CLIP’s performance**. To this end, we generated paraphrases using three different LLMs: GPT-4o-mini (our original setup), GPT-4.1, and Gemini-2.5-Flash. We then measured READ-CLIP’s compositional reasoning performance averaged over five benchmarks used in the main paper.
> > >
> > > | Model | AVG  |
> > > |-|-|
> > > | READ-CLIP (GPT-4o-mini)     | 64.1 |
> > > | READ-CLIP (GPT-4.1)         | 64.3 |
> > > | READ-CLIP (Gemini-2.5-flash)| 64.2 |
> > >
> > > As shown in the table above, using stronger LLMs results in a **slight performance gain**, though the improvement remains **marginal**. This suggests that while paraphrase quality may influence performance, READ-CLIP is already quite robust and does not strongly depend on the specific choice of paraphrase generator.

---

> > > > ### Author Response · Authors · 2025-08-07
> > > >
> > > > ### (2) On the Diversity:
> > > >
> > > > To investigate the impact of diversity in LLM-generated captions in our method on compositional reasoning, we **conducted an additional controlled experiment** as suggested by the reviewer.
> > > >
> > > > In our original method, we generated only one paraphrased caption per original (i.e., `num_p` = 1) within the sentence alignment loss, where `num_p` denotes the number of LLM-generated paraphrases per caption. In this new setup, we **increased the number of paraphrases per caption (`num_p` ∈ {1, 3, 5}) to introduce more diversity**.  During training, we randomly sampled one paraphrase from the set of generated variants at each step, while keeping all other components within the sentence alignment loss unchanged.
> > > >
> > > > | Models | CREPE | SUGARCREPE | VALSE | WHATSUP | SUGARCREPE++ (ITT) | SUGARCREPE++ (TOT) | Average |
> > > > |-|-|-|-|-|-|-|-|
> > > > | READ-CLIP (`num_p` = 1) | **41.5**  | **87.0**         | 76.2  | **43.9**    | 69.8             | 66.2             | **64.1**    |
> > > > | READ-CLIP (`num_p` = 3) | 41.1  | 86.5       | **76.4**  | 43.4    | 70.0             | 66.4             | 64.0    |
> > > > | READ-CLIP (`num_p` = 5) | 41.3  | 86.5       | 76.0  | 43.6    | **70.8**             | **66.6**             | **64.1**    |
> > > >
> > > > As shown in the table above, the performance on SUGARCREPE++ dataset slightly improves as `num_p` increases. On the other hand, increasing `num_p` does not lead to a clear improvement in the performance averaged over the five benchmarks. Recall that the SUGARCREPE++ benchmark specifically requires the model to recognize semantic similarity between paraphrased caption pair. Therefore, this result suggests that greater diversity in paraphrases helps the model better capture the semantic similarity between paraphrased pairs.
> > > >
> > > > Thanks to the reviewer’s insightful suggestion, we were able to more carefully examine the impact of diversity in LLM-generated captions. We believe this analysis can provide useful guidance for future work on paraphrase-aware training strategies. Accordingly, we will include this additional experiment and discussion in the revised manuscript.

---

> > > > > ### Comment · Reviewer_gRGm · 2025-08-08
> > > > >
> > > > > The clarifications and additional results adequately address the points I raised. I will maintain my positive rating.

---

> > > > > > ### Author Response · Authors · 2025-08-08
> > > > > >
> > > > > > We appreciate your response and are happy to hear that your question have been addressed.

---

### Official Review · Reviewer_Zdd4 · 2025-07-02

**Clarity:** 2
**Significance:** 2
**Originality:** 3
**Rating:** 4
**Confidence:** 4

**Summary:**

This paper proposes a fine-tuning approach for vision-language models (VLMs) such as CLIP to enhance their compositional reasoning capabilities. The method introduces two auxiliary loss terms to the standard contrastive loss: (1) a token-level reconstruction loss that encourages the model to reconstruct paraphrased versions of the input sentence using a frozen pre-trained decoder, and (2) a sentence-level alignment objective. This fine-tuning strategy yields improved performance across multiple compositional reasoning benchmarks. Furthermore, integrating this approach with other compositional reasoning techniques leads to additional performance gains.

**Questions:**

1. The paper presents aggregate performance across five benchmarks; however, it would be helpful to understand how the proposed approach performs on specific subsets of these benchmarks. Are there particular subsets where the method shows significant improvement, or conversely, any subsets where performance degrades?
2.According to Appendix A.1, the paraphrases used for the token-level reconstruction loss are selected from the COCO dataset. However, COCO captions are often diverse and may not be true paraphrases, with many describing different semantic content. How did the authors address this issue during training, particularly when the paired captions are not semantically equivalent?
3. When comparing results in Tables 1 and 4, it is unclear why applying READ to the fine-tuned CLIP model (Table 4: average performance of CLIP+READ = 58.2) underperforms compared to applying READ to the pre-trained CLIP model (Table 1: CLIP+READ = 64.1). Could the authors clarify the cause of this performance drop? Is there any explanation for why the fine-tuning process may negatively affect READ's effectiveness?

I am willing to improve my score if the above issues are clarified.

**Ethical Concerns:**

["NO or VERY MINOR ethics concerns only"]

**Final Justification:**

Most of my concerns are resolved by the authors rebuttal. But I would still like to know how training with sentences which are not semantically similar lead to improved performance.

**Limitations:**

yes

**Paper Formatting Concerns:**

There are no major formatting issues

**Quality:**

3

**Strengths And Weaknesses:**

Strengths:
- The paper is well written and easy to follow
- The idea of reconstructing paraphrased versions of input sentences is innovative -- particularly, the use of an off-the-shelf decoder to reconstruct paraphrases encourages the network to capture the syntactic structure and word order of the input text.
- The introduction of a sentence-level alignment loss is another noteworthy contribution. This objective encourages the model to learn semantic representations by bringing semantically equivalent sentences—despite lexical variation—closer in the embedding space.

Weaknesses:
- **Limited evaluation results:** While Tables 1, 2 and 3 present results across 5 different benchmarks, the evaluation on the subsets of the benchmarks is missing. Reporting performance on benchmark subsets would offer deeper insights into which aspects of compositionality and semantics the proposed method improves.
- **Clarification on the Use of "Paraphrase":** The term paraphrase in the token reconstruction loss appears to be misleading. According to Section A.1 of the appendix, these paraphrases are selected from the COCO dataset, which consists of diverse captions that may not always be semantically equivalent. This distinction is not clearly addressed in the paper and may explain why the method performs best for K=1 in Table 3.
- **Reliability of LLM-Generated Paraphrases:** For the sentence-level alignment loss, the paraphrases are generated by the LLM GPT-4o-mini. However, the paper does not discuss the reliability or semantic fidelity of these generated paraphrases, nor does it mention whether any post-processing or filtering was applied. This information is crucial for assessing the robustness of the alignment loss.
- **Performance on other downstream tasks:** It is necessary to show that the proposed approach retains the performance of CLIP on other downstream tasks, while improving the performance on compositionality benchmarks. For example, Table 6 in the NegCLIP paper (https://openreview.net/pdf?id=KRLUvxh8uaX) provides such an analysis, showing performance retention across standard vision and language benchmarks. Including a similar evaluation would help establish the robustness and general applicability of the fine-tuning strategy.

---

> ### Author Rebuttal · Authors · 2025-07-31
>
> ## To Reviewer 3 (Zdd4)
>
>
> We thank the Reviewer Zdd4 for the detailed review and constructive suggestions. We appreciate your acknowledgement that our idea is innovative, the contribution is noteworthy, and the paper is easy to follow. Please find our answers to your comments and questions as follows.
>
>
> **[R3-1]** `The evaluation on the subsets of the benchmarks is missing.`
>
> Below we provide a comprehensive **per-subset performance**. Due to the space limitation, we compare with CLIP and FSC-CLIP (the latest baseline).
>
> **READ-CLIP consistently outperforms CLIP across most subsets**, not just in average scores. Notably:
>
> - On **SugarCrepe and SugarCrepe++**, READ-CLIP shows **up to 20 percentage point improvements** in **replace** and **swap** tasks.
> - On **VALSE**, READ-CLIP has strong gains in **existence** (69.3 → 80.8), **plurals** (57.9 → 73.8), and **relations** (66.0 → 75.0), all of which require understanding fine-grained semantic variation.
> - On **WHATSUP**, READ-CLIP yields improvements across all subcategories.
> - On **CREPE**, significant performance improvement is observed on **Atom** (18.9 → 37.8) and **Swap** (17.3 → 61.2), while performance drop is observed on **Negate**.
>
>
> We will include the complete table as well as the above discussion in the revised manuscript.
>
> ### SugarCrepe
> * att: attribute, obj: object, rel: relation
>
> | Model|Add (att)|Add (obj)|Replace (att)|Replace (obj)|Replace (rel)|Swap (att)|Swap (obj)|Total|
> |-|-|-|-|-|-|-|-|-|
> | CLIP (Pre-trained) | 69.5    | 77.0    | 80.3         | 90.7         | 69.4         | 64.1      | 61.2      | 73.2   |
> | READ-CLIP         | **87.7**    | **90.3**    | **91.0**         | **94.9**         | **80.6**         | **82.7**      | **81.6**      | **87.0**   |
> | FSC-CLIP          | 86.7    | 90.2    | 89.2         | 94.3         | 80.4         | 77.8      | 77.6      | 85.2   |
>
>
> ### SugarCrepe++ (ITT)
>
> |Model|Replace (att) |Replace (obj)|Replace (rel)|Swap (att)|Swap (obj)|Total|
> |-|-|-|-|-|-|-|
> | CLIP (Pre-trained) | 65.7 | 87.0 | 56.5 | 45.0 | 45.8 | 60.0 |
> | READ-CLIP | **72.2** | 90.1 | 57.5 | **66.2**     | **62.9**     | **69.8**  |
> | FSC-CLIP | 73.5 | **90.4** | **60.1** | 60.4  | 55.1 | 67.9 |
>
>
> ### VALSE
>
> | Model | Actions | Coreference | Counting | Existence | Noun Phrases | Plurals | Relations | Total |
> |-|-|-|-|-|-|-|-|-|
> | CLIP (Pre-trained) | 74.3 | 54.4 | 61.7 | 69.3 | 90.4 | 57.9 | 66.0 | 67.4 |
> | READ-CLIP | **86.3** | 55.7 | **69.0** | **80.8** | **95.8** | **73.8** | 75.0 | **76.2** |
> | FSC-CLIP | 82.9 | **59.4** | 66.3 | 77.6      | 93.5 | 72.7 | **75.3** | 74.4 |
>
>
> ### WHATSUP
>
> | Model | COCO-Spatial | GQA-Spatial | Whats-up | Total |
> |-|-|-|-|-|
> | CLIP (Pre-trained) | 44.5 | 47.8 | 30.7    | 41.0  |
> | READ-CLIP         | **51.6** | **48.1** | **31.8**    | **43.9**  |
> | FSC-CLIP          | 47.7 | 41.9 | 29.6    | 39.8  |
>
>
> ### CREPE
>
> | Model | Atom | Negate | Swap  | Average |
> |-|-|-|-|--|
> | CLIP (Pre-trained) | 18.9 | 35.3   | 17.3  | 23.9  |
> | READ-CLIP | 37.1 | 26.2   | **61.2**  | 41.5  |
> | FSC-CLIP | **40.5** | **41.3**   | 45.5  | **42.5** |
>
> ---
>
> **[R3-2]** `The term "paraphrase" in the token reconstruction loss is misleading.`
>
> As the reviewer correctly observes, captions in the COCO dataset describe different aspects of the same image and are not necessarily semantically equivalent. We used such alternative captions for the reconstruction loss.
>
> In the submitted manuscript, we referred to these alternative captions as “paraphrases,” but **this term is misleading** as pointed out by the reviewer. More precisely, our method samples **caption variants**—captions associated with the same image that describe complementary or overlapping content.
>
> In the revised manuscript, **we will revise the terminology** from “paraphrase” to “caption variant” or “alternative caption” to better reflect their nature and avoid confusion.
>
> We also agree with your interpretation that the performance peak at $K = 1$ in Table 3 may stem from this potential semantic difference among captions. We will explicitly include this explanation in the revised manuscript to clarify the notation and interpretation of our experimental result in Table 3.
>
>
> ---
>
> **[R3-3]** `LLM-Generated Paraphrases are Reliable?`
>
> Thank you for raising this important question.
>
> First, we did not apply any post-processing or filtering to the LLM-generated paraphrases used during training. This decision was motivated by recent studies that demonstrate the effectiveness of using LLM-generated captions in contrastive learning pipelines, even without filtering [Fan'23, Doveh’23-1, Doveh’23-2].
>
>
> As per the reviewer's comment, **we ran additional experiments validating the reliability of the LLM-generated captions**. To be specific, we randomly sampled **1,000 paraphrase pairs** (original vs. LLM-generated) from our training set and assessed their semantic consistency using **two stronger LLMs**:
> - **GPT-4.1-2025-04-14**
> - **Gemini-2.5-Flash**
>
> Each model is independently prompted to determine whether the two captions are **semantically equivalent** using a strict boolean format, where the detailed prompt is shown below:
>
> #### Prompt (used in both models):
> ```text
> <task>
> Given two image captions, decide whether they are semantically equivalent (i.e., paraphrases).
>
> Return the result **only** as a strict JSON object with one boolean field named "paraphrase".
>
> Example formats:
> {"paraphrase": True}
> {"paraphrase": False}
>
> No additional text, explanations, or metadata.
> </task>
>
> **Input**:
> Caption 1: {original_caption}
> Caption 2: {synthetic_caption}
> ```
>
> #### Result:
>
> | Evaluator         | % Judged as Paraphrase | Std Dev   |
> |------------------|------------------------|-----------|
> | **GPT-4.1**       | 99.7%                  | 0.0547    |
> | **Gemini-2.5-Flash** | 98.1%                  | 0.1366    |
>
> These results indicate that **more than 98%** of our paraphrases were judged to preserve the original meaning, providing strong empirical support for the semantic reliability of our LLM-generated captions.
>
> To further demonstrate the reliability of LLM-generated captions, we conducted additional experiments analyzing how the choice of different LLMs (GPT-4o-mini, GPT-4.1, Gemini-2.5-flash) for caption generation impacts the compositional reasoning performance of READ-CLIP, averaged out over 5 benchmarks used in the manuscript. As shown in the result below, the performance of READ-CLIP remains consistently stable regardless of which LLM is used to generate captions.
>
> | Model                       | AVG  |
> |-----------------------------|------|
> | READ-CLIP (GPT-4o-mini)     | 64.1 |
> | READ-CLIP (GPT-4.1)         | 64.3 |
> | READ-CLIP (Gemini-2.5-flash)| 64.2 |
>
> We will include this discussion in the revised manuscript.
>
> ***References:***
>
> [Fan'23] Fan et al. Improving CLIP training with language rewrites. NeurIPS 2023.
>
> [Doveh'23-1] Doveh et al. Dense and aligned captions (DAC) promote compositional reasoning in VL models. In NeurIPS 2023.
>
> [Doveh'23-2] Doveh et al. Teaching structured vision & language concepts to vision & language models. In CVPR 2023.
>
> ---
>
> **[R3-4]** `Performance on other downstream tasks?`
>
> As per the reviewer's questions, we ran additional experiments on two downstream tasks -- cross-modal retrieval and image classification.
>
> Due to the space limitation, please check the detailed result in our response to `[R2-2]` -- one can confirm that READ-CLIP maintains or even improves the performance of CLIP on **cross-modal retrieval benchmarks**, while some performance drop is observed on *image classification tasks* when READ is applied to CLIP. We note that this slight degradation on classification tasks is a common trade-off observed in prior work on CLIP-based compositional fine-tuning [Oh'24].
>
> We will report these results in the revised manuscript.
>
> [Oh'24] Oh et al. Preserving multi-modal capabilities of pre-trained VLMs for improving vision-linguistic compositionality. In EMNLP 2024.
>
> ---
>
> **[R3-5]** `Why applying READ to the fine-tuned CLIP model (Table 4: average performance of CLIP+READ = 58.2) underperforms compared to applying READ to the pre-trained CLIP model (Table 1: CLIP+READ = 64.1)?`
>
> Let us clarify the implication of the results in Tables 1 and 4. Let us focus on three methods which are fine-tuned versions (using MS-COCO) of the same pre-trained CLIP model, where different fine-tuning loss is used for each method:
>
> - **Method 0** (Table 1, last row): Uses our full READ loss as in Equation (8), where the contrastive component $\mathcal{L}'_{I \rightarrow T}$ **includes** hard negatives.
>
> - **Method 1** (Table 4, Row 1): Standard CLIP baseline, fine-tuned with only the vanilla contrastive loss (average of losses in Equation (2)), **without** using hard negatives.
>
> - **Method 2** (Table 4, Row 2): Uses the full READ loss from Equation (8), **but** replaces \(\mathcal{L}_{I \rightarrow T}^{**\prime**}\)
> with the standard contrastive loss
> \(\mathcal{L}_{I \rightarrow T}\) in Equation (2), **without** using hard negatives.
>
> As such, the READ-CLIP in Table 1 and the READ-CLIP in Table 4 refer to **two different variants** of our method, where the former includes hard negatives in the loss, while the latter does not. **The performance difference of two methods** (64.1 for method 0 and 58.2 for method 2) **implies the importance of including hard negatives in the fine-tuning loss**.
>
> We acknowledge that this distinction may not have been sufficiently clear in the initial submission and could understandably cause confusion. To address this, we will update the revised manuscript to make these differences clear.

---

> > ### Comment · Reviewer_Zdd4 · 2025-08-05
> > **official comment by Reviewer Zdd4**
> >
> > Thank you very much authors for the comprehensive rebuttal. Most of my concerns are addressed. I have raised my score to 4 (borderline accept). It is still not clear to me how training with sentence pairs that are not semantically same will improve the performance of VLMs.

---

> > > ### Author Response · Authors · 2025-08-05
> > >
> > > Thank you for revisiting our rebuttal and raising the score. We understand that the benefit of training on sentence pairs that are not fully semantically identical warrants clearer justification. Our view is that, while not identical in meaning, these captions often share most of the grounded visual content, which can help the model learn semantic relationships between words in the caption. We are exploring additional experiments and plan to include further discussion to clarify this point in the revision.

---

> > > > ### Comment · Reviewer_Zdd4 · 2025-08-05
> > > > **official comment by Reviewer Zdd4**
> > > >
> > > > Thank the authors for their prompt response. Including further discussion will add value to the paper. I understand the challenges involved in obtaining a large-scale vision-language paraphrased dataset. Despite a few concerns regarding the methodology, I find the proposed approach interesting and innovative. Although I maintain my score of 4 (borderline accept), I am inclined to support the acceptance of the paper.

---

### Official Review · Reviewer_a9tz · 2025-07-02

**Clarity:** 3
**Significance:** 3
**Originality:** 3
**Rating:** 5
**Confidence:** 4

**Summary:**

The paper proposes READ to enhance CLIP-like models (ITM) compositional reasoning by adding two auxiliary losses to the usual image-text contrastive objective via finetuning: 1) token-level reconstruction, where a frozen T5 decoder is forced to reconstruct a paraphrase of the input caption from its text-encoder embedding, encouraging the encoder to encode intra-sentence relations; 2) sentence-level alignment; where contrastive alignment of paraphrase pairs is enforced, pushing their embeddings together while keeping other captions apart. Authors show results on standard benchmarks.

**Questions:**

- How sensitive is READ to paraphrase quality (noisy paraphrases)?

- Does READ hurt zero-shot generalization on standard CLIP tasks (e.g., ImageNet, etc)?

- Why not use the same loss objectives with another text backbone?

**Ethical Concerns:**

["NO or VERY MINOR ethics concerns only"]

**Final Justification:**

The authors have addressed my questions, and I'm increasing my score accordingly.

**Limitations:**

Yes

**Paper Formatting Concerns:**

No concerns.

**Quality:**

3

**Strengths And Weaknesses:**

- This paper is overall easy to read, the method is described really well. It is also well motivated, and figures help understand the whole pipeline. Formulations seem correct, extensive results in standard benchmarks, and good analysis.

- This paper focuses on CLIP-like models, given that VLMs cover both auto-regressive and ITM, it should be good to make the distinction explicitly.

- This paper explores only T5 text encoder. While this is ok, relevant prior work is not further discussed. As an example, VQAScore for Evaluating Text-to-Visual Models [1] explicitly fine-tunes a T5 text encoder and explains the rationale behind this decision. Is this something the authors of READ also leverage, thus the positive results?

- Evaluation only limited to relatively old benchmarks: it is highly encouraged to report results on more recent/more challenging/realistic benchmarks, e.g.: SugarCrepe++, Winoground, ConMe [2].

[1] Lin, Zhiqiu, et al. "Evaluating text-to-visual generation with image-to-text generation." European Conference on Computer Vision. Cham: Springer Nature Switzerland, 2024.

[2] Huang, Irene, et al. "Conme: Rethinking evaluation of compositional reasoning for modern vlms." Advances in Neural Information Processing Systems 37 (2024): 22927-22946.

---

> ### Author Rebuttal · Authors · 2025-07-31
>
> ## To Reviewer 2 (a9tz)
>
> We thank the Reviewer a9tz for the detailed review and constructive suggestions. We appreciate your acknowledgement that the paper is well motivated, easy to read, and analyzed well. Please find our answers to your comments and questions as follows.
>
>
>
> **[R2-1]**: `This paper focuses on CLIP-like models, given that VLMs cover both auto-regressive and ITM, it should be good to make the distinction explicitly.`
>
>
> Thank you for the helpful observation. As the reviewer correctly points out, vision-language models (VLMs) can be broadly categorized into three classes:
>
> 1. **Image–Text Contrastive (ITC) Models** – e.g., CLIP, which learn via contrastive objectives over image–text pairs using dual encoders.
> 2. **Auto-Regressive Generation Models** – which generate textual descriptions or answers conditioned on images.
> 3. **Image–Text Matching (ITM)-Based Models** – which treat alignment as a classification task over paired inputs.
>
> Our work focuses specifically on the **first category (ITC)**, targeting CLIP-style dual-encoder architectures.
>
> We acknowledge that this distinction was **not made sufficiently clear** in the submitted manuscript. We will clarify this distinction in the revised manuscript.
>
> ---
>
> **[R2-2]**: `This paper explores only T5 text encoder. While this is ok, relevant prior work is not further discussed. As an example, VQAScore for Evaluating Text-to-Visual Models explicitly fine-tunes a T5 text encoder and explains the rationale behind this decision. Is this something the authors of READ also leverage, thus the positive results?`
>
> Thank you for raising this important point.
>
> First, we would like to clarify the difference in the role of T5 between VQAScore and READ-CLIP. VQAScore explicitly **fine-tunes** the T5 text decoder, whereas READ-CLIP fine-tunes the dual encoders while making use of a **frozen** T5 text decoder. In our method, the T5 text decoder remains completely frozen throughout training and is only used to provide sentence-level reconstruction supervision for the text encoder.
>
> Our rationale for using T5 is closely tied to the core idea of our work. One of the key motivations of READ is to improve the training of the encoder-only CLIP architecture by introducing a **token-level reconstruction loss**. This, however, requires a high-quality pre-trained text decoder. T5, as a widely adopted encoder-decoder model with a strong pre-trained decoder and cross-attention capabilities, is a natural fit for this purpose. Importantly, it allows us to achieve the reconstruction objective without modifying the CLIP backbone, aligning with our design goal of enhancing alignment in a minimally invasive way.
>
> We will clarify this design choice in the revised manuscript to avoid any potential confusion.
>
> As per the reviewer's suggestion, we will further include relevant prior work.
>
> ---
>
> **[R2-3]**: `Evaluation only limited to relatively old benchmarks: it is highly encouraged to report results on more recent/more challenging/realistic benchmarks, e.g.: SugarCrepe++, Winoground, ConMe.`
>
>
> Thank you for this suggestion.
>
> #### 1. On SugarCrepe++
>
> We would like to note that **SugarCrepe++ is already included in our evaluation**, as shown in **Tables 1–4 and Figures 4–5** of the main paper. This benchmark is central to our compositional evaluation suite.
>
> **READ-CLIP outperforms other baselines on SugarCrepe++**, as shown in Table 1 of the submitted manuscript.
>
>
> #### 2. On Winoground and ConMe
>
> Winoground and ConMe are important benchmarks. However, Winoground **go beyond compositionality alone** and often requires advanced capabilities, as stated in a related work:
>
> > *“..., we show that solving the Winoground task requires not just compositional language understanding, but a host of other abilities like commonsense reasoning or locating small, out-of-focus objects in low-resolution images”*
> > — *Diwan et al. 2022 [Diwan'22]*
>
> Additionally, ConMe is designed to evaluate **large-scale generative VLMs** (e.g., Flamingo, PaLI, GPT-4V) and **is not used as a standard benchmark for studies on contrastive CLIP fine-tuning**.
>
> Recall that our focus was to assess **compositional reasoning** of **vision-language models trained with contrastive learning**. While testing on Winoground/ConMe is an interesting direction for future exploration, we believe their exclusion does not detract from the core objectives of this study.
>
> We hope this clarification addresses the concern, and we thank the reviewer again for the constructive feedback.
>
> ***References***:
>
> [Diwan'22] Diwan, A., Berry, L., Choi, E., Harwath, D., & Mahowald, K. (2022, December). Why is Winoground Hard? Investigating Failures in Visuolinguistic Compositionality. In Proceedings of the 2022 Conference on Empirical Methods in Natural Language Processing (EMNLP) (pp. 2236–2250). Abu Dhabi, United Arab Emirates: Association for Computational Linguistics.
>
> ---
>
> **[R2-4]**: `How sensitive is READ to paraphrase quality (noisy paraphrases)?`
>
>
> Thank you for this insightful question.
>
> We interpret the reviewer’s concern as asking how robust our method is when some of the paraphrases used for sentence-level alignment are not true paraphrases which serve as semantic noise.
>
> To resolve the reviewer's question, we conducted an additional experiment where we injected noise into the paraphrase data by randomly replacing a fraction of paraphrase captions with **unrelated captions from the dataset**. Specifically, we ran experiments with **10%** and **20%** of the paraphrase pairs being replaced with mismatched captions.
>
> The results are shown below:
>
> | Model                    | CREPE | SUGARCREPE | VALSE | WHATSUP | SUGARCREPE++ (ITT) | SUGARCREPE++ (TOT) | AVG  |
> |--------------------------|-------|------------|--------|---------|---------------------|---------------------|------|
> | READ-CLIP                | 41.5  | 87.0       | 76.2   | 43.9    | 69.8                | 66.2                | 64.1 |
> | READ-CLIP (10% Noise)    | 39.0  | 86.9       | 76.0   | 43.6    | 67.1                | 64.7                | 62.9 |
> | READ-CLIP (20% Noise)    | 38.6  | 86.8       | 76.0   | 43.4    | 65.1                | 62.9                | 62.1 |
>
> As expected, increasing noise leads to performance degradation. However, the drop is relatively modest—**only 1.2 to 2.0 points in average score**—indicating that **READ is reasonably robust even in the presence of moderate paraphrase noise**.
>
> We will include this analysis and discussion in the revised manuscript.
>
> ---
>
> **[R2-5]**: `Does READ hurt zero-shot generalization on standard CLIP tasks (e.g., ImageNet, etc)?`
>
>
> Thank you for raising this important question. To address the reviewer's question, we ran additional experiments on two standard CLIP tasks: cross-modal retrieval and image classifications tasks.
>
> From the results shown in the below tables, one can confirm that READ-CLIP maintains or even improves the performance of CLIP on **cross-modal retrieval benchmarks**, while some performance drop is observed on *image classification tasks* when READ is applied to CLIP.
>
> ##### 1. Zero-shot Cross-modal Retrieval (Flickr30K)
>
> | Model                  | Direction | R@1   | R@5   |
> |------------------------|-----------|--------|--------|
> | CLIP (Pre-trained)     | i2t       | 79.3   | 95.0   |
> |                        | t2i       | 58.8   | 83.4   |
> | **READ-CLIP (Fine-tuned)** | i2t   | **79.8** | **95.1** |
> |                        | t2i       | **68.9** | **90.6** |
>
> The above result shows that applying READ improves **Recall@1 (text-to-image)** by **+10.1%**, demonstrating that READ effectively enhances text representations in line with its design goals. Note that READ-CLIP is fine-tuned only on COCO dataset,
> which implies that the improvement on Flickr30K is a strong indicator of READ’s cross-dataset generalization ability.
>
> ##### 2. Zero-shot Image Classification
>
> | Model             | MNIST | CIFAR10 |
> |-------------------|--------|---------|
> | CLIP (Pre-trained)| 48.25  | 89.83   |
> | Triplet-CLIP      | 34.31  | 89.11   |
> | GNM-CLIP          | 54.33  | 88.45   |
> | CE-CLIP           | 40.56  | 85.90   |
> | FSC-CLIP          | 50.38  | 89.08   |
> | **READ-CLIP**     | 47.95  | 87.14   |
>
> The above result shows that applying READ incurs performance drop on image classification tasks. We note that this slight degradation on classification tasks is a common trade-off observed in prior work on CLIP-based compositional fine-tuning.
>
> We will report these results in the revised manuscript.
>
> ---
>
> **[R2-6]**: `Why not use the same loss objectives with another text backbone?`
>
>
> Thank you for this thoughtful question. We believe your question refers to our choice of using CLIP encoders (backbone), and we would like to share our thoughts in more detail below.
>
> Although we applied READ to vanilla CLIP backbone in the submitted manuscript, our proposed objectives are also applicable to **any vision-language model (VLM) that uses the dual-encoder architecture**, as the reviewer pointed out. These VLMs include a variety of off-the-shelf CLIP variants with different text encoders or backbone modifications.
>
> We will revise the manuscript to explicitly state that **READ can be directly applied to other pre-trained dual-encoder models** in a plug-and-play fashion during fine-tuning.

---

> ### Comment · Area_Chair_huKn · 2025-08-05
>
> Reviewer a9tz, please engage in the discussion period. I understand that the review period started over a weekend, but we only have a few days remaining in the (now slightly extended) discussion period. The author's have provided a thoughtful response to your review and you are obligated to respond to it. You should share with the authors if they addressed questions or concerns you had, and seek clarification about any questions or concerns that remain. Please post your response as soon as you can so that there is time for the authors to follow up and discussion to progress as needed.

---

> ### Comment · Reviewer_a9tz · 2025-08-05
>
> First, I would like to thank the authors for their responses. However, now I have more questions and concerns, which I summarize below (my apologies for a bit late response, I had to re-read the paper and re-read prior works):
>
> - *T5, as a widely adopted encoder-decoder model with a strong pre-trained decoder and cross-attention capabilities, is a natural fit for this purpose.*: I would like the authors to further expand on this. As far as I know, OpenAI CLIP, EVA-CLIP uses a GPT-2 style decoder transformer; ALIGN and SigLIP uses a BERT-base encoder, OpenCLIP uses a RoBERTa-base encoder, M-CLIP uses a XML-R encoder. On the other hand, generative models like SimVL, CoCa, BLIP-2 leverage a T5-like encoder-decoder.
>
> - Furthermore, VQAScore authors find that "FlanT5 benefits from bidirectional image-question encoding and extensive training on challenging QA datasets."; this brings again the question of not using other text backbones that does not benefit from *bidirectional image-question encoding and extensive training on challenging QA dataset*.
>
> - T5 is pre-trained to reconstruct arbitrary spans that have been masked out, so the encoder learns bidirectional context while the decoder learns generative semantics; this seems to give the main strength in the proposed approach, not the proposed READ method. However, without empirical evidence, it is hard to say whether this is true or not. Using a different text backbone could have easily clarify this.
>
> - Winoground results: this dataset is particularly small and easy to run; results should be available within minutes. It is particularly unfortunate that Diwan et al. 2022 [Diwan'22] was able to avoid testing on this dataset, since this particularity *"commonsense reasoning or locating small, out-of-focus objects in low-resolution images"* are in fact, some of the core principles for compositionality and realistic settings.
>
> - It also seems evident that READ hurt zero-shot generalization on standard CLIP tasks, typically, results on 21 datasets are reported. These results are important, and the evaluation of zero-shot generalization after fine-tuning yields important insights about the task, showing both effectiveness and robustness of the proposed method.
>
> - Taking a closer look, and thanks to the discussion with other reviewers (Zdd4), prior work has also explored using paraphrases or extensions of the textual inputs generated by other models ([Doveh’23-1]), however, in the Related Work section, it seems the authors refer to this work as *"approaches typically generate hard negative captions"* (L87-88). This is not accurate, since some of the cited work generate both expansions/paraphrases, as well as contrastive captions.
>
> I would like the authors to further clarify on these questions before adjusting my final score.

---

> > ### Author Response · Authors · 2025-08-06
> >
> > We sincerely appreciate the reviewer’s thoughtful feedback and follow-up question. Your detailed comment has helped us better understand your questions, and we are currently preparing a more comprehensive response.
> >
> > We would first like to clarify a potential point of confusion about T5 model.
> >
> > In our proposed method, we do not use the full T5 encoder-decoder architecture. Instead, as illustrated in Figure 1, we adopt the CLIP text encoder as our backbone and utilize a frozen T5 decoder solely for token-level caption reconstruction.
> > Therefore, in our setup, the “text backbone” refers to the CLIP encoder, not T5.
> > Furthermore, we use the original pre-trained T5 model, not Flan-T5, which is further fine-tuned on challenging QA datasets.
> >
> > To address your question more thoroughly, we are planning to run additional experiments.
> > Before doing so, we would greatly appreciate your clarification on the following:
> > * Are you specifically asking for a comparison between different decoder architectures (e.g., T5 vs. BART)?
> > * Or are you suggesting replacing the CLIP text encoder with the T5 encoder, i.e., adopting a generative encoder-decoder pipeline instead of the contrastive CLIP setup?
> > * In the latter case, would it be acceptable to retain the CLIP image encoder, even though it is not aligned with the T5 encoder?
> >
> > We look forward to your guidance so we can ensure our additional experiments directly address your intention.

---

> > > ### Comment · Reviewer_a9tz · 2025-08-06
> > >
> > > Oh I see my confusion, thank you so much for re-clarifying; I see no need to replace the T5 decoder for the token-level reconstruction. With that clear, I would really like to see Winoground results and clarification on Related Work section (L87-88).
> > >
> > > Overall, I have no further questions or concerns, and I'm increasing my score accordingly.

---

> > > > ### Author Response · Authors · 2025-08-07
> > > >
> > > > **[R2-6]**: `Results on Winoground`
> > > >
> > > > To address the reviewer’s follow-up question, we conducted an additional evaluation on the **Winoground** benchmark. We compared **READ-CLIP** with the original **CLIP (pre-trained)** and six representative compositional reasoning fine-tuning methods. The performance of random guessing is also reported for reference.
> > > >
> > > > | Model             | Text Score | Image Score | Group Score |
> > > > |------------------|------------|-------------|-------------|
> > > > | Random Guessing     | 25.00      | 25.00       | **16.67**   |
> > > > | CLIP (pre-trained) | 11.25      | 31.25       | 9.00        |
> > > > | GNM-CLIP          | 12.50      | 35.25       | 10.00       |
> > > > | FSC-CLIP          | 11.75      | 32.75       | 9.25        |
> > > > | DAC-SAM           | 12.75      | 21.00       | 8.50        |
> > > > | Triplet-CLIP      | 11.00      | 31.50       | 8.25        |
> > > > | NEG-CLIP          | 10.50      | 30.50       | 8.00        |
> > > > | **READ-CLIP**     | 10.25      | 24.25       | 6.00        |
> > > > | CE-CLIP           | 12.00      | 19.75       | 5.25        |
> > > > | DAC-LLM           | 10.50      | 23.00       | 4.75        |
> > > >
> > > > As shown in the table above, all models achieve **group scores below the performance of random guessing**, despite being designed to improve compositional reasoning. This finding may be attributed in part to the analysis by [Diwan'22], which argued that success on Winoground requires not only compositionality, but also other capabilities including **common-sense reasoning or world knowledge** and **locating small, out-of-focus objects in low-resolution images**.
> > > >
> > > > The above result implies that solving Winoground remains a significant challenge for existing methods that solely focus on the compositional reasoning problem. We will state such limitation in the revised manuscript.
> > > >
> > > > ---
> > > > **[R2-7]**: `Results on Zero-shot Image Classification Tasks`
> > > >
> > > > To address the reviewer’s follow-up question, we conducted **additional evaluations on 23 widely used zero-shot classification benchmarks**. We compared **READ-CLIP** with the original CLIP (pre-trained) and six representative compositional reasoning fine-tuning methods.
> > > >
> > > > |Model|caltech101| cars | cifar10 | cifar100 | country211 | dtd | eurosat | fer2013 | fgvc_aircraft | flowers | food101 | gtsrb | imagenet-o | imagenet1k | imagenet_sketch | imagenetv2 | kitti | mnist | pcam | renderedsst2 | resisc45 | stl10 | voc2007 | avg. |
> > > > |-|-|-|-|-|-|-|-|-|-|-|-|-|-|-|-|-|-|-|-|-|-|-|-|-|
> > > > | CLIP (Pre-trained) | 81.45 | 59.73 | 89.83 | 64.26 | 17.24 | 44.26 | 50.50 | 41.21 | 19.71 | 66.37 | 84.00 | 32.54 | 47.55 | 63.36 | 42.25 | 55.74 | 27.14 | 48.25 | 62.26 | 58.76 | 53.63 | 97.12 | 76.47 | **55.81** |
> > > > | NegCLIP | 82.55 | 53.87 | 88.91 | 62.98 | 15.00 | 42.98 | 49.69 | 46.74 | 16.83 | 64.99 | 79.43 | 30.22 | 46.45 | 60.94 | 40.43 | 53.24 | 27.71 | 49.69 | 54.86 | 58.59 | 52.89 | 96.74 | 79.63 | 54.58 |
> > > > | GNM-CLIP | 81.46 | 53.09 | 88.45 | 65.02 | 15.22 | 42.07 | 50.67 | 45.96 | 17.22 | 63.33 | 81.81 | 30.24 | 47.40 | 61.42 | 40.97 | 54.06 | 25.32 | 54.33 | 55.63 | 58.48 | 49.83 | 96.37 | 77.40 | 54.60 |
> > > > | FSC-CLIP | 81.77 | 51.76 | 89.08 | 64.87 | 14.45 | 40.69 | 51.63 | 49.53 | 15.78 | 61.70 | 78.70 | 29.78 | 45.50 | 59.18 | 38.90 | 51.74 | 29.40 | 50.38 | 50.95 | 59.75 | 52.75 | 96.10 | 79.03 | 54.06 |
> > > > | DAC-LLM | 77.68 | 39.44 | 90.40 | 63.88 | 14.27 | 38.99 | 52.26 | 50.49 | 11.31 | 54.63 | 74.16 | 24.16 | 45.45 | 51.04 | 35.23 | 45.03 | 16.60 | 42.16 | 50.02 | 54.42 | 49.56 | 97.05 | 77.92 | 50.27 |
> > > > | DAC-SAM | 75.65 | 39.90 | 89.88 | 63.71 | 14.78 | 40.00 | 51.15 | 47.65 | 9.00 | 53.91 | 72.30 | 24.85 | 45.50 | 52.35 | 35.07 | 46.81 | 18.71 | 45.33 | 50.02 | 59.80 | 51.67 | 96.10 | 65.77 | 50.00 |
> > > > | **READ-CLIP** | 78.24 | 39.58 | 87.14 | 57.78 | 10.17 | 34.95 | 39.24 | 41.03 | 13.05 | 52.17 | 71.60 | 26.71 | 44.45 | 51.53 | 32.91 | 45.29 | 30.52 | 47.95 | 47.34 | 52.28 | 44.33 | 95.24 | 78.88 | 48.80 |
> > > > | CE-CLIP | 78.34 | 35.28 | 85.90 | 60.07 | 9.53 | 35.21 | 42.83 | 39.52 | 10.02 | 48.19 | 70.12 | 27.98 | 44.80 | 49.91 | 31.52 | 43.23 | 34.60 | 40.56 | 50.04 | 61.23 | 47.65 | 95.75 | 77.30 | 48.68 |
> > > > | Triplet-CLIP | 80.61 | 23.85 | 89.11 | 61.52 | 7.11 | 39.26 | 35.17 | 47.69 | 12.69 | 54.64 | 76.28 | 24.70 | 42.75 | 54.83 | 37.02 | 48.41 | 15.33 | 34.31 | 49.58 | 51.84 | 54.67 | 94.61 | 72.90 | 48.21 |
> > > >
> > > > As shown in the table above, all fine-tuned models, which are designed to improve compositional reasoning, achieve **lower average performance than the original CLIP (pre-trained)** across the 23 datasets.
> > > >
> > > > This result reflects a broader trend observed in prior work [Oh'24], where improvements in compositional understanding often come at the cost of general zero-shot capability. Such trade-offs highlight the challenge of maintaining broad generalization while specializing models for compositional reasoning.
> > > >
> > > > We will include this result to show the limitation of our method (as well as existing methods for handling compositional reasoning problem) in the revised manuscript.

---

> > > > > ### Author Response · Authors · 2025-08-07
> > > > >
> > > > > **[R2-8]**: `Clarification on Related Work Section`
> > > > >
> > > > > We thank the reviewer for the thoughtful comment and for pointing this out. In our original description in the *Related Work* section, we referred prior works collectively as:
> > > > >
> > > > > > “These approaches typically generate hard negative captions [5,6,20,54,62,65,69], or construct hard negative pairs by altering both text and image [3, 41, 50, 88].”
> > > > >
> > > > > We now realize that this phrasing may have unintentionally overlooked distinctions between different methods. We acknowledge that providing a more fine-grained and accurate categorization is essential for guiding future research in this area. To that end, we plan to revise the paragraph to reflect these distinctions more clearly. Specifically, we will update it to:
> > > > >
> > > > > > “These approaches typically generate hard negative captions via rule-based perturbation [5, 65], language models [6, 69], scene graphs [20, 54], or construct hard negative pairs by altering both text and image [3, 41, 50, 88]. Among these, DAC [5] highlights the importance of training with dense and well-aligned captions for improving compositional reasoning, while TSLVC [6] finds that using paraphrased captions in analogy loss improves zero-shot image classification performance.”
> > > > >
> > > > > This revision better reflects the range of methods used and makes it clear that [Doveh’23-2] (which is [6] in the above paragraph) also incorporates **paraphrased** captions. We appreciate the reviewer’s suggestion and will make sure to include this change in the revised manuscript.
> > > > >
> > > > > ***References***:
> > > > >
> > > > > [Diwan'22] Diwan, A., Berry, L., Choi, E., Harwath, D., & Mahowald, K. (2022, December). Why is Winoground hard? Investigating failures in visuolinguistic compositionality. In Y. Goldberg, Z. Kozareva, & Y. Zhang (Eds.), Proceedings of the 2022 Conference on Empirical Methods in Natural Language Processing (EMNLP 2022)
> > > > >
> > > > > [Oh'24] Oh et al. Preserving multi-modal capabilities of pre-trained VLMs for improving vision-linguistic compositionality. In EMNLP 2024.
> > > > >
> > > > > [Doveh'23-2] Doveh, S., Arbelle, A., Harary, S., Panda, R., Herzig, R., Schwartz, E., Kim, D., Giryes, R., Feris, R., Ullman, S., & Karlinsky, L. (2023). Teaching structured vision & language concepts to vision & language models. In Proceedings of the IEEE/CVF Conference on Computer Vision and Pattern Recognition (CVPR 2023).

---

> > > > > ### Comment · Reviewer_a9tz · 2025-08-08
> > > > >
> > > > > Thank you again for answering all remaining questions, and for including all revisions and additional results! I don't have further questions (and have updated the final score to reflect full-support for acceptance).

---

> > > > > > ### Author Response · Authors · 2025-08-09
> > > > > >
> > > > > > We sincerely appreciate the score increase and are pleased that our responses have addressed your questions!

---

### Official Review · Reviewer_UjDy · 2025-07-03

**Clarity:** 3
**Significance:** 3
**Originality:** 3
**Rating:** 4
**Confidence:** 4

**Summary:**

In this paper, the authors propose a novel fine-tuning framework for CLIP-like VLM, including reconstruction and alignment of text descriptions to enhance the text encoder. Extensive experiments have demonstrated that the proposed algorithm can significantly enhance compositional reasoning capabilities.

**Questions:**

Besides the above concerns raised by the reviewer, please answer the following questions:

1. Please provide additional details regarding the paraphrased captions. How are the captions generated, and which tool is used for generating them?
2. If the granularity of the captions is further enhanced, would the performance improve accordingly?

**Ethical Concerns:**

["NO or VERY MINOR ethics concerns only"]

**Limitations:**

Yes

**Quality:**

3

**Strengths And Weaknesses:**

Strengths:
The paper is well-written, with clear logic and comprehensive experiments. The proposed algorithm is reasonable and has been validated as effective.

Weekness:
1. Although the technical details differ, the proposed algorithm shares a similar motivation with two existing CLIP-based papers, FG-CLIP[1] and FineCLIP[2]. Both of these aim to enhance the fine-grained representation capability of the CLIP model by leveraging fine-grained image-level and region-level multimodal contrastive learning. Please compare the proposed algorithm with these works in terms of algorithmic details and performance (if possible).
2. The training dataset used in this study is COCO, which contains many images featuring similar objects and collected from similar scenes. How can it be ensured that fake negative pairs do not adversely affect the model's training?

[1] FG-CLIP: Fine-Grained Visual and Textual Alignment
[2] FineCLIP: Self-distilled Region-based CLIP for Better Fine-grained Understanding

---

> ### Author Rebuttal · Authors · 2025-07-31
>
> ## To Reviewer 1 (UjDy)
>
> We thank the Reviewer UjDy for the detailed review and constructive suggestions. We appreciate your acknowledgement that the paper is well-written, with clear logic and comprehensive experiments. Please find our answers to your comments and questions as follows.
>
>
> **[R1-1]**: `Although the technical details differ, the proposed algorithm shares a similar motivation with two existing CLIP-based papers, FG-CLIP and FineCLIP. Both of these aim to enhance the fine-grained representation capability of the CLIP model by leveraging fine-grained image-level and region-level multimodal contrastive learning. Please compare the proposed algorithm with these works in terms of algorithmic details and performance (if possible)`
>
>
> We appreciate the reviewer’s thoughtful feedback. While **READ shares the high-level motivation** of improving CLIP's fine-grained compositional understanding, our approach is **distinct in scope, algorithmic design, and empirical setting**. Below, we clarify these differences in depth and provide **quantitative comparisons** to support our claims.
>
>
> #### **1. Overall Distinction**
>
> - **FG-CLIP** and **FineCLIP** enhance **visual representations** by explicitly aligning image regions and parts with fine-grained textual cues using large-scale region-level annotation or dynamic distillation strategies.
> - In contrast, **READ introduces lightweight, text-centric auxiliary losses** to structurally regularize the **text encoder**:
>   - *(i)* a **token-level reconstruction loss**, and
>   - *(ii)* a **sentence-level alignment loss** using LLM-generated paraphrases.
>
> The main differences between FG-CLIP [Xie'25], FineCLIP [Jing'24], and READ-CLIP are summarized in the table below:
>
> `[Table R1]` Conceptual Differences of READ-CLIP and baselines
> | Method      | Targeted Modality | Key Mechanisms                                     | Training Data Scale                                 |
> |-------------|-------------------|----------------------------------------------------|---------------------------------------------|
> | **FG-CLIP** | Image encoder     | Region-level contrastive alignment, hard negatives | 1.6B global + 12M images / 40M boxes         |
> | **FineCLIP**| Image encoder     | Multi-grained contrastive + self-distillation      | MS-COCO (100K) + 970K region-labels              |
> | **READ** *(Ours)* | Text encoder      | Token reconstruction + sentence alignment           | MS-COCO (100K)              |
>
>
> #### **2. Performance Comparison on Compositional Reasoning Benchmarks**
>
> `[Table R2]` Performance Comparison between READ-CLIP and baselines
>
> | Model                   | CREPE | SUGARCREPE | VALSE | WHATSUP |
> |-------------------------|-------|------------|-------|---------|
> | FG-CLIP (ViT-B/16)      | 15.7  | 80.4       | 69.6  | 40.7    |
> | FineCLIP (ViT-B/16)     | 14.2  | 80.7       | 67.8  | 40.8    |
> | **READ-CLIP (ViT-B/32)**| **41.5** | **87.0**   | **76.2** | **43.9** |
>
> `Table R2` shows that, even with the **coarse granularity (i.e., large patch size and lower spatial resolution) of ViT-B/32 relative to ViT-B/16**  and **modest training data scale** (as shown in `Table R1`), **READ-CLIP consistently outperforms FG-CLIP and FineCLIP across all compositional reasoning benchmarks**, highlighting the effectiveness of its text-encoder-centric approach for enhancing compositional reasoning capability in CLIP.
>
> We hope this clarifies the detailed difference between READ, FG-CLIP and FineCLIP, and provides a clear message that READ is a simple and effective alternative.
>
> ***References:***
>
> [Xie'25] Xie, C., Wang, B., Kong, F., Li, J., Liang, D., Zhang, G., Leng, D., & Yin, Y. (2025). FG-CLIP: Fine-grained visual and textual alignment. In Proceedings of the 42nd International Conference on Machine Learning (ICML 2025).
>
> [Jing'24] Jing, D., He, X., Luo, Y., Fei, N., Yang, G., Wei, W., Zhao, H., & Lu, Z. (2024). FineCLIP: Self-distilled region-based CLIP for better fine-grained understanding. In Advances in Neural Information Processing Systems, 37 (NeurIPS 2024).
>
> ---
>
> **[R1-2]**: `The training dataset used in this study is COCO, which contains many images featuring similar objects and collected from similar scenes. How can it be ensured that fake negative pairs do not adversely affect the model's training?`
>
>
> Thank you for raising this important concern. We fully agree that the **semantic false negative problem**—i.e., negative pairs that are semantically similar to positives—is a key challenge in contrastive learning.
>
> Nonetheless, **MS-COCO remains the standard for CLIP-style contrastive fine-tuning**, and is widely used not only in recent compositional representation learning studies such as NegCLIP [Yuksekgonul'23] and FSC-CLIP [Oh'24], but also in recent multi-modal representation learning such as M^3-Mix [Oh'23] and CLIP-Refine [Yamaguchi'23]. These prior works have shown that, **with COCO datasets, contrastive objectives leads to meaningful performance gains**, indicating that the practical effect of false negatives is limited in this setting. Thus, we believe that our experimental setup is both **standardized and empirically sound**.
>
> We appreciate the reviewer’s thoughtful question and hope this clarifies our position.
>
> ***References:***
>
> [Yuksekgonul'23] Yuksekgonul, M., Bianchi, F., Kalluri, P., Jurafsky, D., & Zou, J. (2023). When and why vision-language models behave like bags-of-words, and what to do about it? Proceedings of the 11th International Conference on Learning Representations (ICLR 2023).
>
> [Oh'24] Oh, Y., Cho, J. W., Kim, D.-J., Kweon, I. S., & Kim, J. (2024). Preserving multi-modal capabilities of pre-trained VLMs for improving vision-linguistic compositionality. Proceedings of the 2024 Conference on Empirical Methods in Natural Language Processing (EMNLP 2024).
>
> [Oh'23] Oh, C., So, J., Byun, H., Lim, Y., Shin, M., Jeon, J.-J., & Song, K. (2023). Geodesic multi-modal mixup for robust fine-tuning. Proceedings of the 37th Conference on Neural Information Processing Systems (NeurIPS 2023).
>
> [Yamaguchi'23] Yamaguchi, S., Feng, D., Kanai, S., Adachi, K., & Chijiwa, D. (2025). Post-pre-training for modality alignment in vision-language foundation models. Proceedings of the IEEE/CVF Conference on Computer Vision and Pattern Recognition (CVPR 2025).
>
> ---
>
>
> **[R1-3]**: `Please provide additional details regarding the paraphrased captions. How are the captions generated, and which tool is used for generating them?`
>
> Thank you for your question regarding the generation process of paraphrased captions used in our method.
>
> As described in **Appendix A.1** of the supplementary material, the **sentence-level alignment loss** in READ requires semantically equivalent yet lexically distinct captions for each image–caption pair. To obtain these, we generate paraphrases using a structured LLM-based procedure:
>
> - **Model**: `gpt-4o-mini-2024-07-18` via the OpenAI API
> - **Prompt format**: Shown in *Appendix Figure 6*
> - **Generation parameters**: `temperature = 1.0` (all other settings default)
> - **Timing**: Paraphrases are generated *offline*, prior to training
> - **Quantity**: One paraphrase per original caption
>
> These synthetic captions are used **only** for supervising the sentence-level alignment loss.
> For the **token-level reconstruction loss**, we instead leverage the *multiple original captions* provided by MS-COCO, without any LLM-based augmentation. These details are in the Appendix of the submitted manuscript.
>
> We will include these details in the main body of the revised manuscript.
>
>
> ---
>
> **[R1-4]**: `If the granularity of the captions is further enhanced, would the performance improve accordingly?`
>
>
> Thank you for your insightful question. We agree that the **granularity of captions can affect model performance**. Indeed, prior work on compositional reasoning—such as FSC-CLIP [Oh'24], DAC [Doveh'23-1], and TSLVC [Doveh'23-2]—has demonstrated that replacing generic captions from datasets like CC3M with **fine-grained, VLM-generated descriptions** can lead to consistent improvements.
>
> In our case, READ is designed to promote understanding relationship between words through both reconstruction and alignment objectives. As such, we believe that **incorporating more fine-grained captions would further enrich the training signal and potentially enhance model performance**.
>
> We appreciate this valuable suggestion and will incorporate a discussion on the potential impact of caption granularity on READ's performance in the revised manuscript.
>
> ***References:***
>
> [Oh'24] Oh, Y., Cho, J. W., Kim, D.-J., Kweon, I. S., & Kim, J. (2024). Preserving multi-modal capabilities of pre-trained VLMs for improving vision-linguistic compositionality. Proceedings of the 2024 Conference on Empirical Methods in Natural Language Processing (EMNLP 2024).
>
> [Doveh'23-1] Doveh, S., Arbelle, A., Harary, S., Herzig, R., Kim, D., Cascante-Bonilla, P., Alfassy, A., Panda, R., Giryes, R., Feris, R., Ullman, S., & Karlinsky, L. (2023). Dense and aligned captions (DAC) promote compositional reasoning in VL models. In Advances in Neural Information Processing Systems, 37 (NeurIPS 2023 Spotlight).
>
> [Doveh'23-2] Doveh, S., Arbelle, A., Harary, S., Panda, R., Herzig, R., Schwartz, E., Kim, D., Giryes, R., Feris, R., Ullman, S., & Karlinsky, L. (2023). Teaching structured vision & language concepts to vision & language models. In Proceedings of the IEEE/CVF Conference on Computer Vision and Pattern Recognition (CVPR 2023).

---

> ### Comment · Area_Chair_huKn · 2025-08-05
>
> Reviewer UjDy, please engage in the discussion period. I understand that the review period started over a weekend, but we only have a few days remaining in the (now slightly extended) discussion period. The author's have provided a thoughtful response to your review and you are obligated to respond to it. You should share with the authors if they addressed questions or concerns you had, and seek clarification about any questions or concerns that remain. Please post your response as soon as you can so that there is time for the authors to follow up and discussion to progress as needed.

---

### Decision · Program_Chairs · 2025-09-17

**Decision:**

Accept (poster)

**Comment:**

The paper proposes a fine-tuning method for CLIP-like vision-language models that enhances compositional reasoning by incorporating two auxiliary objectives (a token-level reconstruction, which encourages the text encoder to capture intra-sentence structure by reconstructing a paraphrased caption, and a sentence-level alignment, which pulls paraphrase pairs together in the embedding space while pushing apart unrelated captions). Across the reviews, reviewers were positive about the paper being clearly written and well-motivated, and thought the proposed objectives were novel, technically sound and effective. They also noted the strong empirical performance across a variety of benchmarks.

Initially there were some concerns raised including limited evaluation on more recent benchmarks, lack of generalization analysis across diverse backbones, and concerns about the reliability of paraphrase generation. However, these were largely addressed in the rebuttal and discussion, with the authors providing extended experiments (e.g., on Winoground and zero-shot tasks), clarifying distinctions in related work, justifying design choices (e.g., T5 usage), and committing to camera-ready updates. For the camera-ready version, the authors should be to incorporate these important elements from the rebuttal/discussion (especially clarifications on paraphrase quality and terminology, updated evaluation results on Winoground and zero-shot tasks, and improved exposition of related work distinctions).

The AC recommends acceptance.